# Review of Bumpless Build Cube (BBCube) Using Wafer-on-Wafer (WOW) and Chip-on-Wafer (COW) for Tera-Scale Three-Dimensional Integration (3DI)

**Takayuki Ohba [1,\*], Koji Sakui [1], Shinji Sugatani [1], Hiroyuki Ryoson [1,2] and Norio Chujo [1,3]**

1    Institute of Innovative Research, Tokyo Institute of Technology, 4259 Nagatsuta, Midori-ku, Yokohama 226-8503, Japan; sakui.k.aa@m.titech.ac.jp (K.S.); sugatani.s.aa@m.titech.ac.jp (S.S.); hiroyuki.ryoson@dexerials.com (H.R.); norio.chujo.fj@hitachi.com (N.C.)
2    Dexerials Co.,Tochigi 323-0194, Japan
3    Hitachi, Ltd., Tokyo 185-8601, Japan
\*    Correspondence: ohba.t.ac@m.titech.ac.jp

**Abstract:** Bumpless Build Cube (BBCube) using Wafer-on-Wafer (WOW) and Chip-on-Wafer (COW) for Tera-Scale Three-Dimensional Integration (3DI) is discussed. Bumpless interconnects between wafers and between chips and wafers are a second-generation alternative to the use of micro-bumps for WOW and COW technologies. WOW and COW technologies for BBCube can be used for homogeneous and heterogeneous 3DI, respectively. Ultra-thinning of wafers down to 4 μm offers the advantage of a small form factor, not only in terms of the total volume of 3D ICs, but also the aspect ratio of Through-Silicon-Vias (TSVs). Bumpless interconnect technology can increase the number of TSVs per chip due to the finer TSV pitch and the lower impedance of bumpless TSV interconnects. In addition, high-density TSV interconnects with a short length provide the highest thermal dissipation from high-temperature devices such as CPUs and GPUs. This paper describes the process platform for BBCube WOW and COW technologies and BBCube DRAMs with high speed and low IO buffer power by enhancing parallelism and increasing yield by using a vertically replaceable memory block architecture, and also presents a comparison of thermal characteristics in 3D structures constructed with micro-bumps and BBCube.

**Keywords:** bumpless; TSV; WOW; COW; BBCube; bandwidth; yield; power consumption; thermal management

## 1. Introduction

Semiconductor devices and computer systems have evolved as feature sizes have been continuously reduced [1–4]. On the other hand, three-dimensional technology has been considered since the 1980s, mainly from the viewpoint of monolithic ICs [5–11]. From the late 1990s, 3D technology has been widely studied for the hybrid structure, including package from the die-level to wafer-level, e.g., how to stack semiconductor elements and how to connect between stacked dies with the vertical interconnects such as TSVs [12–27].

According to this trend, computer system volumes will reach 50 mm$^3$, and the power consumption will be 0.5 mW [28,29]. Even in such small computers, high performance and large memory capacity are desired without sacrificing power efficiency and thermal dissipation. Conventional two-dimensional (2D) scaling and three-dimensional (3D) integration methods such as those used in High-Bandwidth-Memory (HBM) [30], however, will inevitably face an economic crisis due to the manufacturing costs and yield required [31–33].

A promising approach to overcome these problems is to combine 3D stacking with high throughput, i.e., co-integration extended into the third dimension (z-direction) using Wafer-on-Wafer (WOW) and Chip-on-Wafer (COW) technologies. In detail, the z-height of a multi-wafer stack must be small, meaning that there should be no bumps between dies, and

the dies should be thin. This is the main feature of BBCube, which allows high bandwidth with low power consumption because of the short length of TSVs and high-density signal parallelism [34]. Furthermore, high-density TSVs act as thermal pipes, and, hence, a low temperature, even in a 3D structure, can be expected.

## 2. Manufacturing Cost Crisis for Two-Dimensional Scaling

Before discussing 3D integration for high-volume manufacturing, it is necessary to investigate the current status and future prospects of semiconductor technology development. Conventional 2D scaling will face a severe economic crisis due to the expensive lithography processes and facilities required. Reducing costs requires the adoption of advanced lithography technologies, which, together with peripheral support facilities such as a defect monitoring system, account for one-third to one-fourth of the total cost of a manufacturing line. Furthermore, bit cost is saturated around 20 nm nodes [35,36] due to unavoidable invisible defect reduction. Unless there is sufficient yield, the total cost will increase even if high-resolution lithography is employed. This is the main reason why multiple, small microprocessor dies (chiplets) are integrated [37,38]. In short, while useful for reducing chip size, scaling is extremely burdensome in terms of capital investment. Large-scale investments to the new fabrication facilities (Fabs) have so far been made considering the technologies that will be available two to three generations ahead without any major technology changes. This is based on the empirical rule in the semiconductor that profits are made several generations after investments for reasons involving the trade-offs between products sales and facility depreciation.

According to this empirical rule, an investment in recently developed 7 nm technology needs to be made in consideration of its applicability to 2–3 nm technologies. In the case of ArF ($\lambda$ = 193 nm), immersion lithography, double or quad patterning for one layer is needed to meet to those critical pattern dimensions. Extreme ultraviolet (EUV; $\lambda$ = 13.5 nm) lithography has the potential to allow patterning in a single step, and thus EUV is superior to ArF. However, the price of EUV lithography machines is more than 120 million USD [39], which is more than twice that of ArF immersion (iArF) lithography machines, and their current throughput is less than that of iArF machines. When converted into the processing capacity of current large-scale Fabs (e.g., 50,000 incoming wafers per month), based on this system performance, an investment of about 2 billion USD will be required for EUV technology. Assuming that the lifelong sales for each generation are about 10-times the corresponding business investment, the corresponding market size necessary for this investment is more than 20 billion USD. Although, this estimate is based on the 440 billion USD total worldwide semiconductor sales in 2020, this market size for one product and one manufacturer is not realistic.

In conclusion, this is one of the limits of two-dimensional scaling in light of the economics of the industry, and it is difficult to find a scenario of victory at present, especially beyond nanometer node.

## 3. Paradigm Shift to Bumpless Build Cube Integration

Extending structures into vertical space (z-direction), for example, by three-dimensional stacking, in combination with conventional two-dimensional integration, is anticipated to overcome the problems noted above. The concept of Bumpless Build Cube (BBCube) is a solution to the problems of next-generation 2.5D (side-by-side arrays) and 3D stack systems, in which device dies and interposers are connected without bumps, described in Section 8.3.

Figure 1 shows a comparison of the bump and bumpless interconnects using TSVs, assuming eight dies for a memory core and one logic controller. Since a chip-level stack formed by Chip-on-Chip (COC) technology using bump connections needs pick-and-place for chip transfer, the die thickness is limited by the mechanical stiffness requirements and warpage, resulting in a chip pitch of around 80–100 μm. The mechanical stiffness decreases with die thickness [40,41].

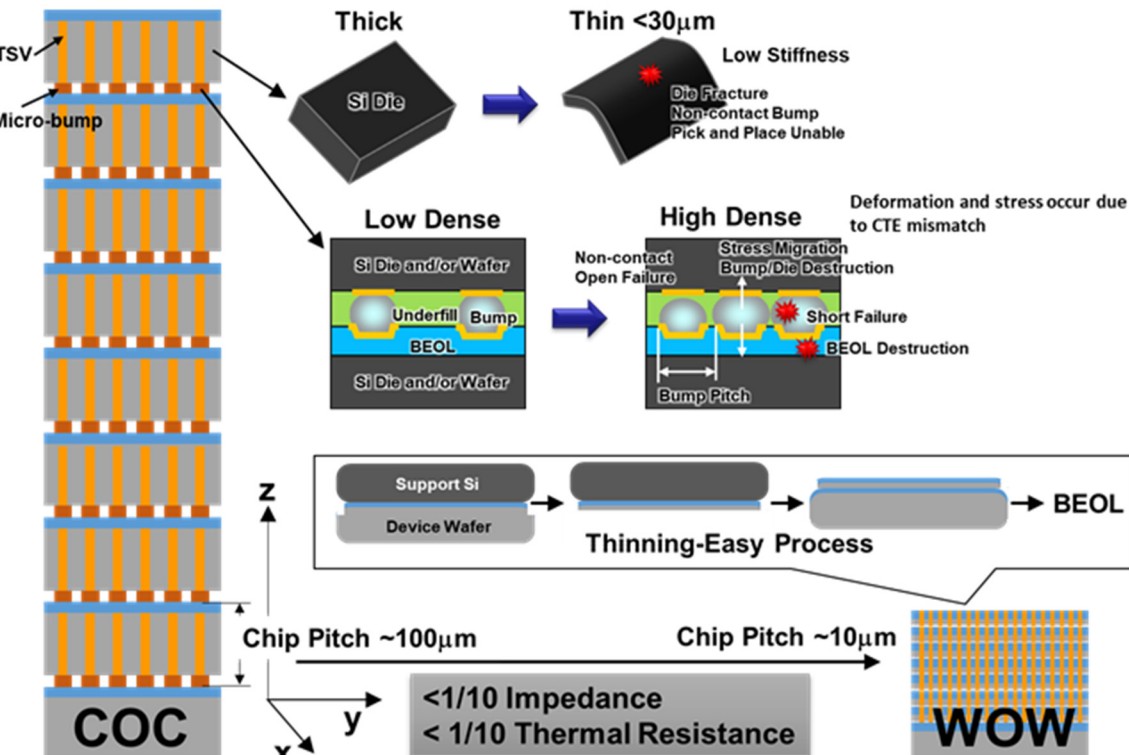

**Figure 1.** A comparison of bump and bumpless interconnects using TSVs for 3D logic/memory stack structures, assuming eight dies for a memory stack and one logic controller. Since the thickness of the die and density of bumps are limited by mechanical and process difficulties, chip pitch becomes as large as approximately 100 μm, in accordance with the die thickness and bump height. Shortened bumpless interconnects can be formed with higher density (narrower pitch) compared with TSVs and bumps due to the limitations of bump size and pitch. By using wafer thinning and bumpless interconnects, the chip pitch becomes about 1/10, and impedance and thermal resistance become less than 1/10 due to the absence of bumps characterized by high electrical and thermal resistance.

If the bump height varies, some bumps will not come into contact with the electrodes on the chip surface. When high pressure is applied to avoid such contact failures, bonding failures due to plastic deformation are mitigated. However, if excessive pressure is applied, problems such as electrical shorts and destruction will occur between bumps due to the lateral deformation of bumps and the multi-interconnects under bumps due to vertical concentrated stress [42–44]. These problems become more significant when the bump pitch is narrowed. This limits the density of TSVs that can be achieved using a combination of TSVs and bumps.

The WOW process consists of *bonding-first* using a thinned wafer and then the formation of TSV interconnects. Thus, the wafer thickness is determined by whether thinning degrades the device characteristics. There was no damage when a DRAM Si wafer was thinned to 4 μm [45–48]. The wafer (chip) stack pitch was around 10 μm, which is 1/10th as thick as that of COC. The WOW process enabled wafer thinning from 775 μm to 1 μm, as shown in Figure 2. For thinning of a DRAM wafer, the effects of Si thickness, the thinning method, and Cu contamination from the backside on the device characteristics of 20 nm-node DRAMs were evaluated [49]. No obvious degradation of the retention characteristics occurred, even when the Si thickness was reduced to 3 μm, as shown in Figure 3. The refresh time was improved by increasing the thickness of the backside defects layer using grinding. The backside defects act as a trapping site for the Cu diffusion and thus Cu diffusion is prevented when the backside has sufficient defects. From the perspective of reliability, due to the poor gettering ability of the CMP finished surface, is necessary to optimize the gettering ability if there is concern about Cu contamination during the process.

This suggests that it is important to design the diffusion length of defects carefully to prevent defects entering the depletion layer, taking the standby currents and the retention characteristics into account, as shown in Figure 4.

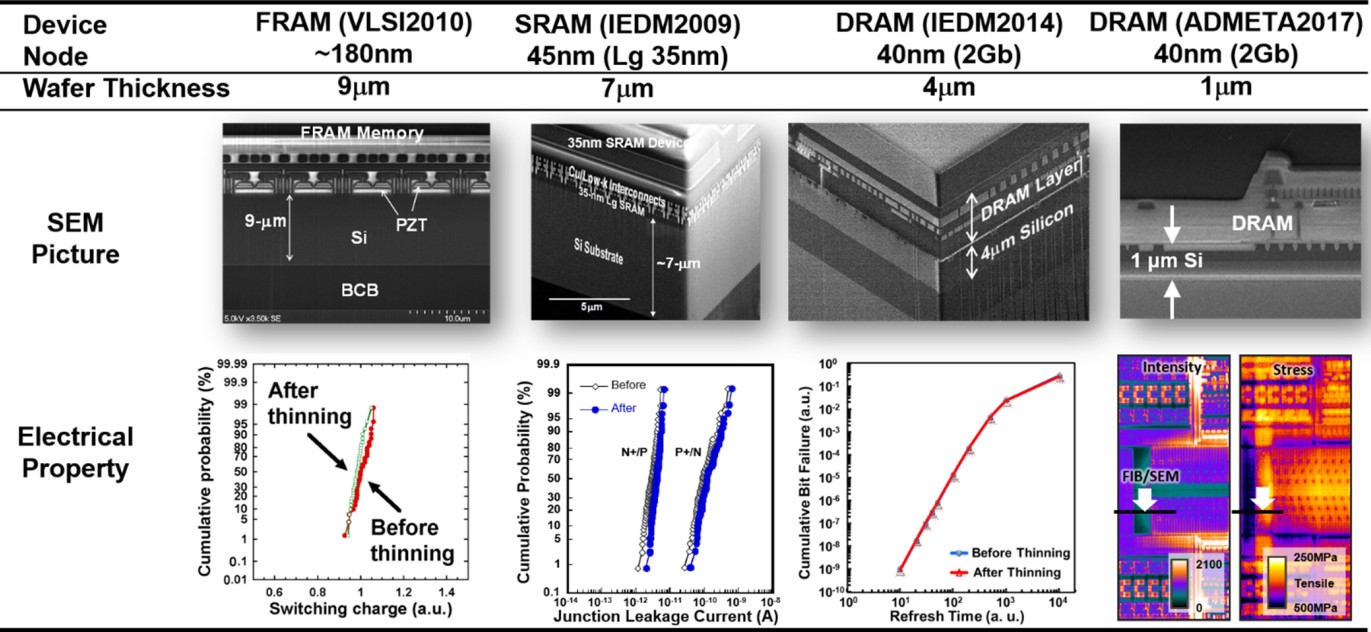

**Figure 2.** Cross-sectional SEM images and electrical properties after device wafer thinning. Thinning was carried out from 9 to 1 μm for FRAM, SRAM, and DRAMs, respectively. There was no degradation in the electrical characteristics after thinning, and the circuit area at the critical layer could be observed from the back side of the wafer when the silicon thickness became one micrometer.

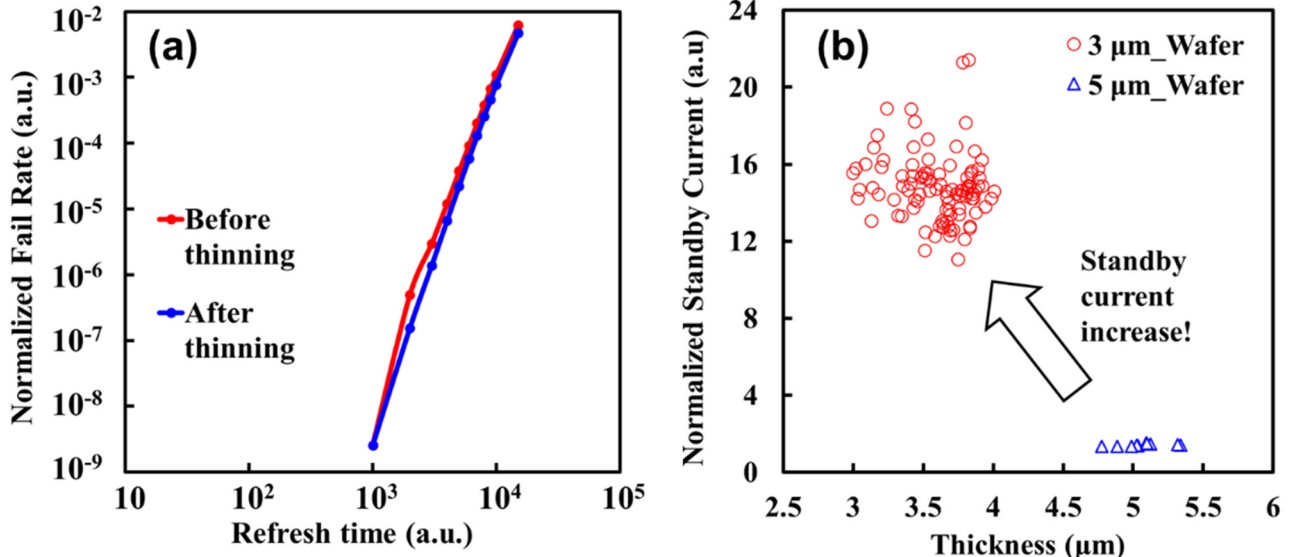

**Figure 3.** (**a**) Comparison of retention time distributions of same chip before and after fine grinding (#2000 grit abrasive) to 3 μm followed by Cu contamination at ~$10^{14}$/cm$^2$, which is 1000-times higher than that of the BEOL process (<$10^{11}$ atoms/cm$^2$); (**b**) Standby current as a function of Si thickness by fine grinding with and without Cu contamination. Standby current were measured from 3 μm (100 points) and 5 μm (10 points) wafers.

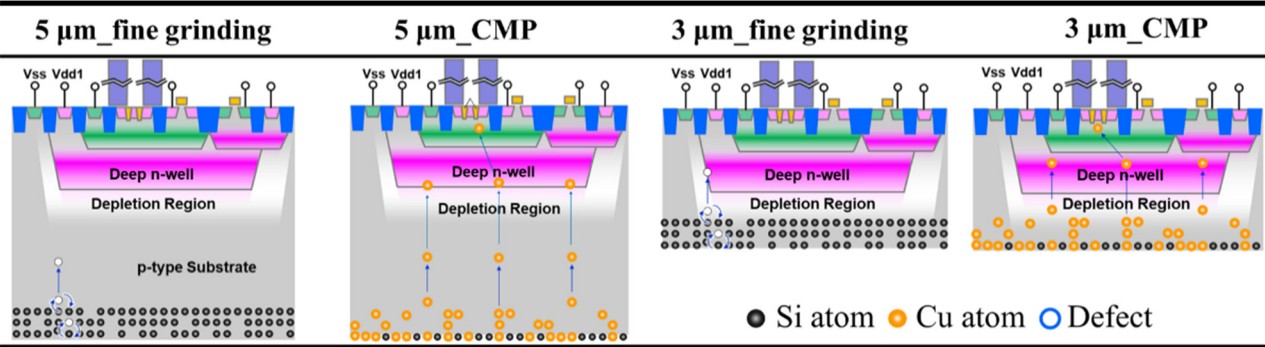

**Figure 4.** Schematic diagram of cross-sectional image of transistor and the degradation model for DRAMs after thinning. The depth of a deep N-well is about 2.0–3.0 μm. The depth of the depletion region between the deep N-well and the substrate is calculated to be 3.0–4.0 μm. In the case of a Si thickness of 5.0 μm, the defects do not reach the depletion region, and thus both the standby current and retention characteristics do not change. When the Si thickness is being reduced to 3 μm by fine grinding, the defects reach the depletion region. These defects in the depletion region increase the junction leakage current between the substrate (Vss) and the deep N-well (Vdd1). This causes an increase in standby current. On the other hand, since CMP treatment removes the grinding-induced defects and reduces diffused-defects at the depletion region, the standby current is improved.

Since the physical length of TSV interconnects is determined by the wafer thickness, including the device layer and adhesive, the total length in the case of an eight-wafer stack was <80 μm. Trends of TSV interconnects versus the number of stacked chips and/or wafers were estimated as shown in Figure 5 [50–52]. The total height, based on the die-to-die pitch, was less than 0.5 mm, even for a stack of 60 wafers. The TSV density ranged from $10^6/cm^2$ to $10^7/cm^2$, which is 10- to 100-times larger than the case of TSVs and bump interconnects.

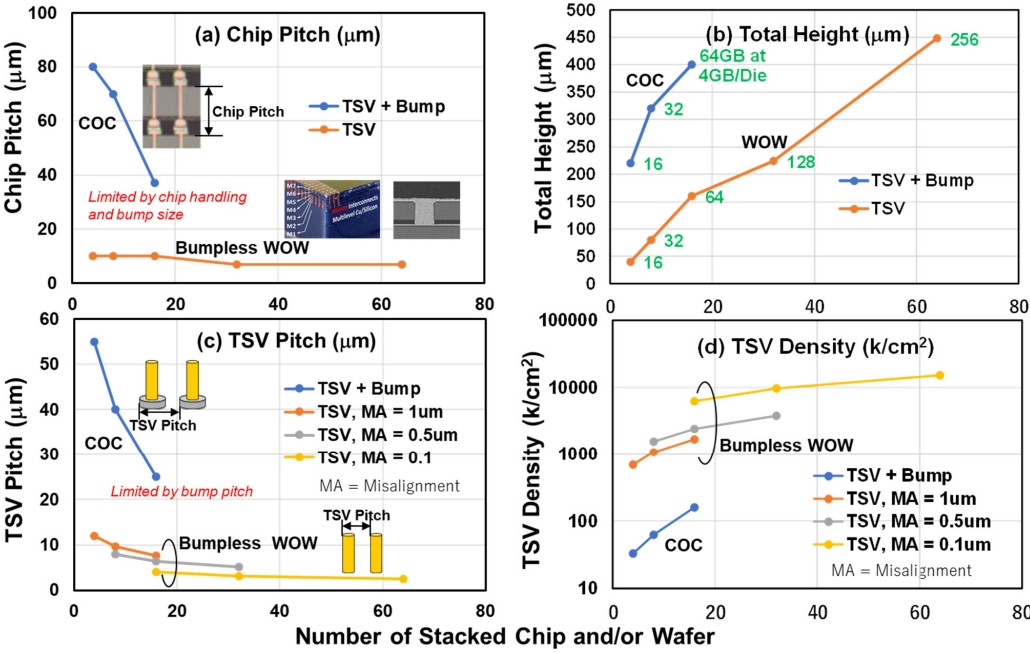

*COC/COW Specification of TSV + Bump: Nick Kim, "Digest of the 2018 IEDM Short Course, 2018; Marc Loranger, SW Test Workshop 2016.*

**Figure 5.** Trends of TSV interconnects as a function of number of stacked layers: (**a**) chip pitch, (**b**) total height, (**c**) TSV pitch, and (**d**) TSV density. For bumpless process, TSV diameter and space between pads were 9 to 1 μm and 1 μm, varied with misalignment (MA), respectively.

It was possible to next make a roadmap to achieve a high-bandwidth system and high-density integration backed up by production costs. Moreover, retaining Cu interconnects technology and the standard 300 mm wafer size for stacking ensures compatibility with existing manufacturing facilities in Front-End processing and helps utilize the mature process technologies that have been developed for wafer processing.

Since the wiring length of the TSVs is determined by the thickness of the wafer, the wiring length becomes shorter when the wafer is thinned down. The conventional wiring length consists of the length of Cu wiring used for TSVs and the bump height, and the total length is about 80–100 μm. The resistance of the bumps is about one order of magnitude higher than that of Cu, e.g., Sn-3.5Ag (12.3 μΩcm) >> Cu (1.68 μΩcm). If only TSVs are used, and there are no bumps with high resistance and the wiring resistance is reduced to <1/10 at a length of 10 μm and a constant diameter. Because of the high density and low resistance of TSVs, high bandwidth and low power consumption can be expected. Details will be discussed in Section 8.

Bumpless Build Cube (BBCube) is a second-generation alternative to the use of TSVs with micro-bumps. The BBCube, bumpless interconnects process involves a *Thinning-First* process before bonding wafers, followed by a *Via-Last* process, meaning that interconnects are formed after bonding the wafers, as shown in Figures 6 and 7. Via-hole etching was carried out, followed by lithography, on a silicon substrate with multilevel interconnects and a device layer after bonding the thinned wafer. Since bumpless Wafer-on-Wafer (WOW) technology uses a back-to-front stack, in principle, any number of thinned 300 mm wafers can be stacked to fabricate large-capacity memory and logic devices. This wafer stacking method is similar to multilevel metallization in the Back-End-of-Line (BEOL), as if replacing dielectric deposition using thinned· wafers and Al and/or Cu metallization with bumpless interconnects using TSVs.

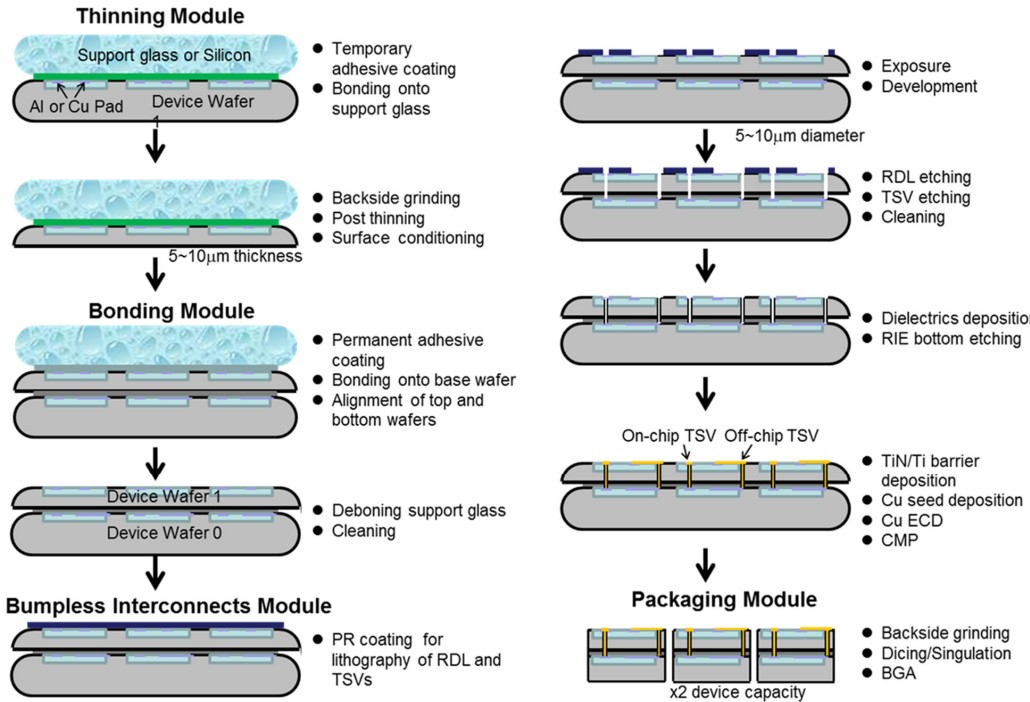

**Figure 6.** Process flow of bumpless interconnects using TSVs and Wafer-on-Wafer (WOW). Additional wafers can be stacked on top without any limitation on the number of wafers. These modules can also be applied to Chip-on-Wafer (COW) after wafer-level molding. On-chip and off-chip TSV, respectively represent bumpless interconnects formed in the device area and the area around devices, including gap fill (molding) materials in COW.

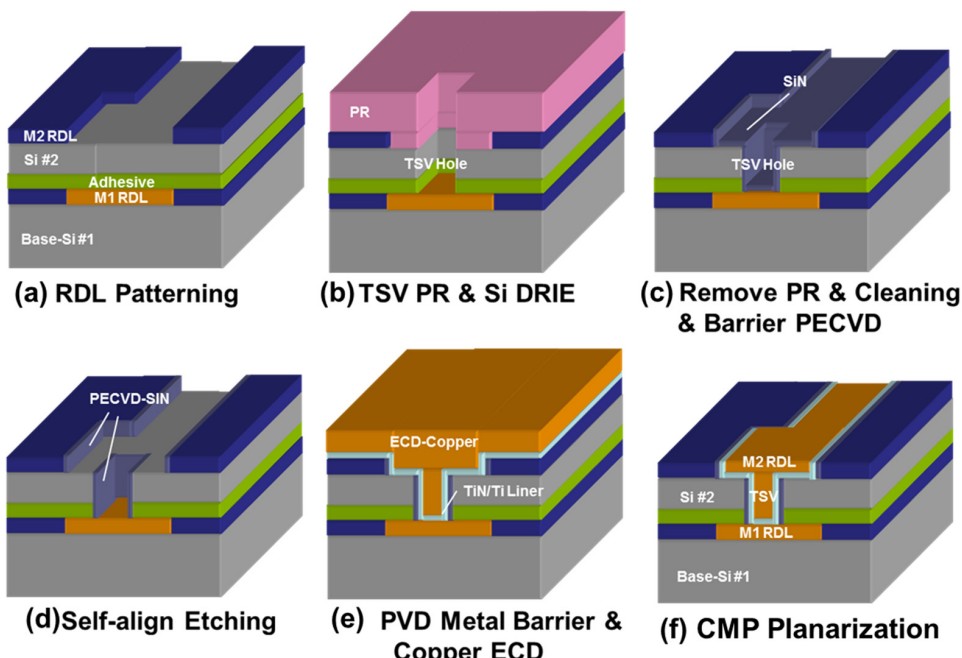

**Figure 7.** RDL formation and TSV (Cu plug) processes. After bonding of thinned wafer to another wafer surface, (**a**). RDL patterning, (**b**). TSV etching, (**c**). barrier layer formation, (**d**). contact opening, (**e**). Cu plug formation by ECD, and (**f**). planarization by CMP is carried out.

In the case of a chip stack for comparison, a singulation step was needed before stacking. There are several methods for singulating a wafer. For example, in the method of forming a dicing groove on the wafer surface in advance, the wafer surface is attached to a film (DAF: Die-attach Film) and the wafer is thinned with a grinder from the backside. This results in singulation by grinding the back surface to the dicing groove. Each of these singulated chips are picked up by a transfer machine and placed on the surface of a separately prepared wafer. These transfer processes are so called pick-and-place. If the chip thickness is small, the rigidity becomes small, and since the chip transfer method is mechanical, it is easy to break the chip. In addition, stress in the device layer generates chip warpage, causing picking errors. Therefore, in the case of COC and COW, a chip thickness of about 20 µm to 30 µm, which satisfies the requirements of the transfer process and mechanical strength, was used. This is the root cause of the thickness limitation in the chip stacking process. The throughput of pick-and-place was obviously low compared to that of WOW, and there was a trade-off between speed and placing accuracy.

The bumpless WOW process proceeded through the development of four modules, classified along the process flow. The modules included (i) a thinning module for thinning the wafer substrates in which devices are implemented, (ii) a stacking module for bonding and stacking with alignment of the wafers, (iii) a TSV interconnects module for forming Cu interconnects embedded in upper and lower wafers with TSVs, and (iv) a packaging module for singulating the stacked wafers. The TSV interconnects module follows the Dual-Damascene process, forms a so-called redistribution layer (RDL) and vertical interconnects simultaneously, and also serves as a counter electrode for the subsequent stacked wafer.

The thickness of the thinned wafer is a critical dimension for the aspect ratio (depth-to-diameter ratio) of TSVs because the aspect ratio is determined by the diameter and the wafer thickness. Since, in this WOW process, a thinned wafer was bonded on a base wafer, there was no need to take measures for handling ultrathin wafers. The typical Si thickness of a thinned wafer is 4 to 5 µm. When the thicknesses of the device layers in a DRAM and an MPU were assumed to be approximately 5 µm and 10 µm, respectively, the aspect ratio of a TSV was only 5 at maximum for a TSV diameter at 3 µm, whereas conventional TSVs with bumps have aspect ratios more than 10 at a die thickness of 30 µm, including

the device layer. With the decreasing aspect ratio, in the TSV processes such as via hole etching, thin-film deposition, and metal filling, the process time decreased to about 1/2 at most, and step coverage significantly improved.

## 4. Details of BBCube WOW Processes

### 4.1. Thinning Module

According to the process flow in Figure 6, a wafer with a device layer was bonded to a support substrate (glass or Si wafer) from the device surface with a temporary adhesive in advance. Thinning was performed by mechanical grinding from the back surface of the wafer (Back Grind, or BG) to within several micrometers of the target thickness, followed by polishing until the final thickness was achieved. The final silicon thickness is the thickness at which no degradation of the device characteristics occurs. This was demonstrated with a DRAM device, which is highly sensitive to defects and metal contamination at the diffusion region [53]. The temporary adhesive and the support substrate were removed after thinning the wafer. A permanent adhesive layer with a thickness of 1 to 5 μm was used for wafer stacking [54,55]. The thickness of the permanent adhesive layer can be reduced according to the surface topography of the device wafer, such as the presence of multilevel interconnects and dicing lines.

The reason for using a support substrate is that if the wafer is made thin, it loses its rigidity and bends under its own weight, making it difficult to handle in the wafer process. This can be more intuitively understood by considering that we could not easily handle thin aluminum kitchen foil even at a thickness of ~12 μm. Wafer thinning was carried out from the back side with a grinder using a grinding wheel. In order to grind from the initial thickness of 775 μm to the micrometer level, taking throughput and the wafer flatness into account, two different sizes of abrasive grains, 50 μm and <5 μm were used one after another for high-speed grinding and low-speed grinding with surface conditioning, respectively. When the abrasive grain size is reduced, large defects generated at the wafer surface can be removed [56,57].

The thickness variation of the thinned wafer was determined by the geometric parallelism between the surface of the grinder and the wafer surface. Because of the mechanism of surface grinding, there was a very small angle between the grinding wheel surface and the wafer surface, so the outer thickness of wafer was slightly smaller than the center thickness. On the other hand, the thinned wafer deformed according to the thickness variation and elastic deformation of the temporary adhesive layer as the rigidity of the wafer decreases. Although the Young's modulus of a silicon wafer in the <110> direction remained the same at about 170 GPa down to 10 μm [58], deformation occurred according to the mechanical properties of the adhesive layer and its thickness. Therefore, the total thickness variation (TTV) within the wafer in a micrometer-level thin region is determined by the variation in the thickness of the temporary adhesive layer for the ground surface.

In the optimized grinding method, when the average thickness was 4 μm, which is just 0.5% of the initial thickness, the total thickness variation (TTV) in the 300 mm wafer was about 1 μm, as shown in Figure 8 [47]. Although the thickness of a wafer consists of the thicknesses of the Si substrate and the transistor layer, including multilevel interconnects, the wafer thickness described here is that of the Si substrate. Since the thickness of the transistor layer is 7 to 15 μm for state-of-the-art DRAM and MPU products, the thickness of the transistor layer becomes predominant when the Si substrate reaches the micron level. In the case of such "silicon thin films," the die-level surface geometry of the device layer, steps at dicing lines, IO pads, and particles affect the local thickness variation of the silicon thin film, when the adhesive layer is not thick enough to absorb the surface geometry and particles.

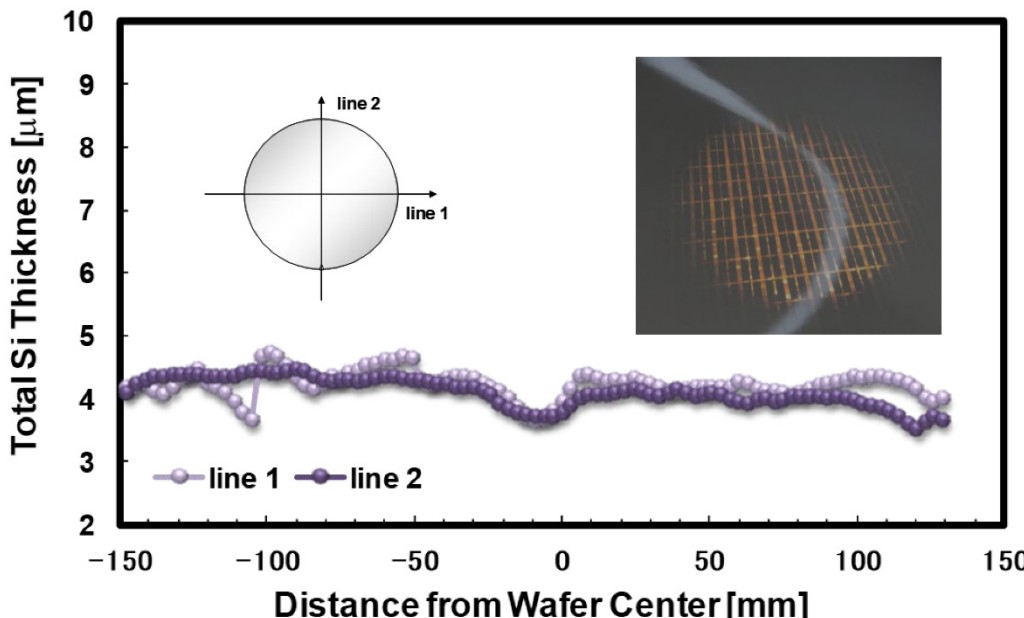

**Figure 8.** Total thickness of 300 mm DRAM wafer after thinning to 4 μm as a function of wafer position from bottom (notch) to top and from left to right. Light transparency occurs at 4 μm where dark region represents high-density device area.

Since the wafer edge was prepared in a bevel shape, a knife edge shape was unavoidably formed when the thinned wafer surface reached the bevel at the inverse taper angle. It is for this reason that edge fractures and cracking tend to occur during the grinding process. To prevent this, a region of about 0.5 to 2 mm from the edge of the wafer was ground to form a step shape before thinning. This process is called edge trimming [59,60]. However, excess edge trimming over 2 mm of edge exclusion in the Front-End process removes some of the usable device area, which leads to a smaller number of dies within the wafers in a multilevel wafer stack. A novel bevel profile for wafer-level multi stacking technology was therefore proposed by considering the relationship between bevel cracking and bevel angle in wafer thinning, using a grinding process [61]. The bevel angle of the wafer was controlled to 45° to 135°, and bevel cracking after grinding was evaluated with a microscope, as shown in Figure 9. When the bevel angle is smaller than 50°, cracks are noticeably generated during thinning by grinding. According to this result, the bevel profile had a bevel angle of 50° for the area used as the device area after thinning, and a region with a bevel angle of 20° to 30° for the area removed by thinning. This bevel design does not need edge trimming and is expected to reduce wafer area loss without the occurrence of cracking during thinning and wafer transportation.

### 4.2. Stacking Module

For WOW stacking, the wafers were aligned using alignment marks on the top and bottom wafers just before being attached and permanently bonded. To ensure alignment, infrared light passing through the silicon substrate was used. The wafers bonded to one another in WOW were thin and thus highly transmissive of light. It is necessary to keep a low coefficient of thermal expansion (CTE) mismatch in the stacking to achieve fine pitch alignment and to reduce wafer warpage. When the temperature of one wafer differs from that of another, the two wafer sizes, which are nominally 300 mm, vary due to CTE; for example, at a temperature difference of only 10 °C, the maximum wafer size difference is 11.7 μm, assuming that the CTE of silicon is $3.9 \times 10^{-6}$/K. Thus, isothermal heating and warpage-free wafers were needed for submicron-level fine-pitch TSV alignment.

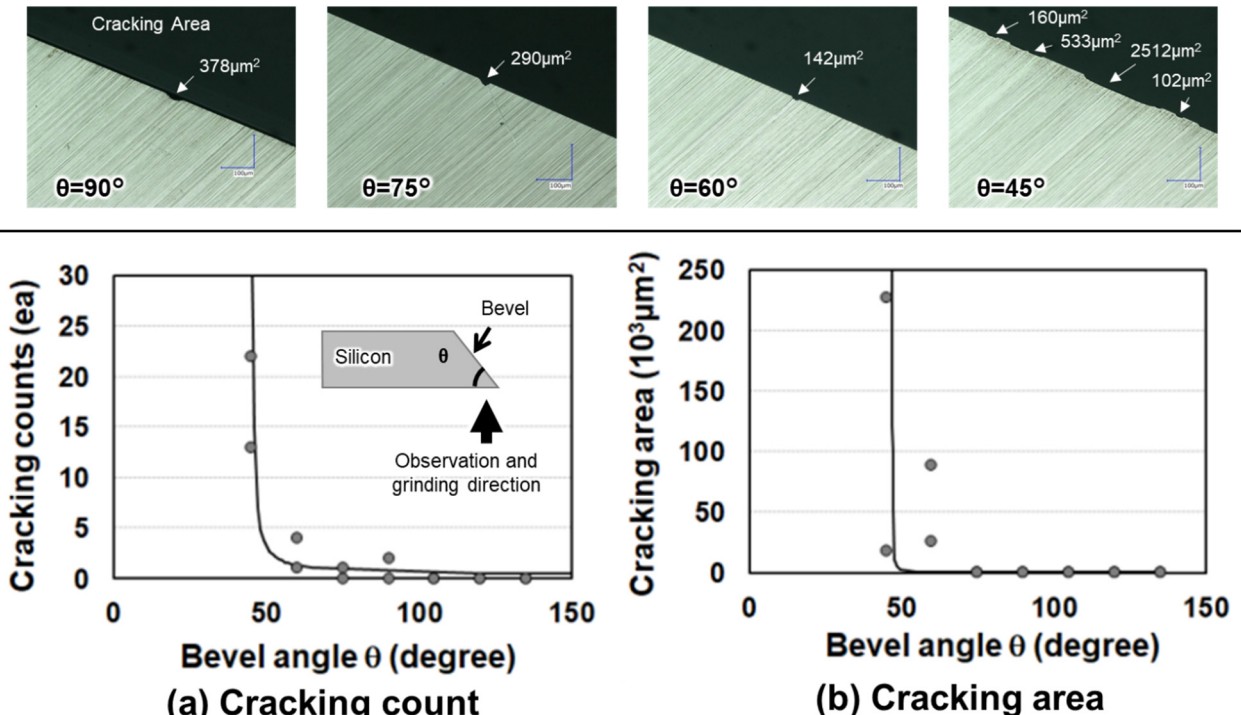

**Figure 9.** Top-view pictures of the wafer edge after grinding for the bevel angle-controlled sample. (**a**,**b**) show the number of cracks and the cracking area as a function of edge position and bevel angle, respectively. Cracking tends to increase with decreasing bevel angle, significantly below 60 degrees. Since cracking occurs randomly along the wafer edge, it might be caused by the grinding step and conditions of the grindstone.

In general, important issues related to the wafer stack process with alignment are thermal stability of materials during the wafer stacking process and matching of the operating temperatures of the temporary and permanent adhesives. Temporary adhesives are de-bondable by heat, UV light, and/or mechanical force and are useful in conventional stacking methods used to fabricate both COW and WOW. To de-bond a device wafer from a support substrate with low stress, a heat de-bondable adhesive called a hot-melt adhesive, with a wide range of operating temperatures is required. Most of the compounds for permanent adhesives with high thermal stability, such as benzocyclobutene (BCB) resins (curing temperature, 250 °C) [62], require high temperatures for the curing process. On the other hand, low-temperature curable compounds generally have poor thermal stability. One candidate is a reactive hot-melt type temporary adhesive (DTB-TP005, Daicel Co., Tokyo, Japan) and permanent adhesive (DPAS100, Daicel Co.) consisting of an organic-inorganic hybrid structure [63]. Figure 10 shows the experimental setup and the glass transition temperature (Tg) of the temporary adhesive controlled at the bonding and de-bonding temperatures. The device wafer and carrier wafer were bonded from the device surface at a temperature of around 130 °C with a temporary adhesive layer of less than 10 μm in thickness formed by a spin-on technique. The device wafer, which was fixed to the carrier wafer with a temporary adhesive layer, was thinned to about 10 μm by grinding and polishing using DGP8761HC (DISCO Corp., Tokyo, Japan), and then coated with an adhesion promoter containing dual functionality in the molecular structure. The adhesion promoter and permanent adhesive layer were sequentially formed in this order on the surface of another device wafer. In the next process, the coated device wafer was stacked on a thinned wafer coated with an adhesion promoter. The thickness of the permanent adhesive layer was about 2.5 μm. After a curing process, there were no voids between the two stacked wafers. The carrier was de-bonded by mechanical peeling-off at 80 kPa using

a differential pressure de-bonder, and then the residual temporary adhesive was able to be removed with an organic solvent.

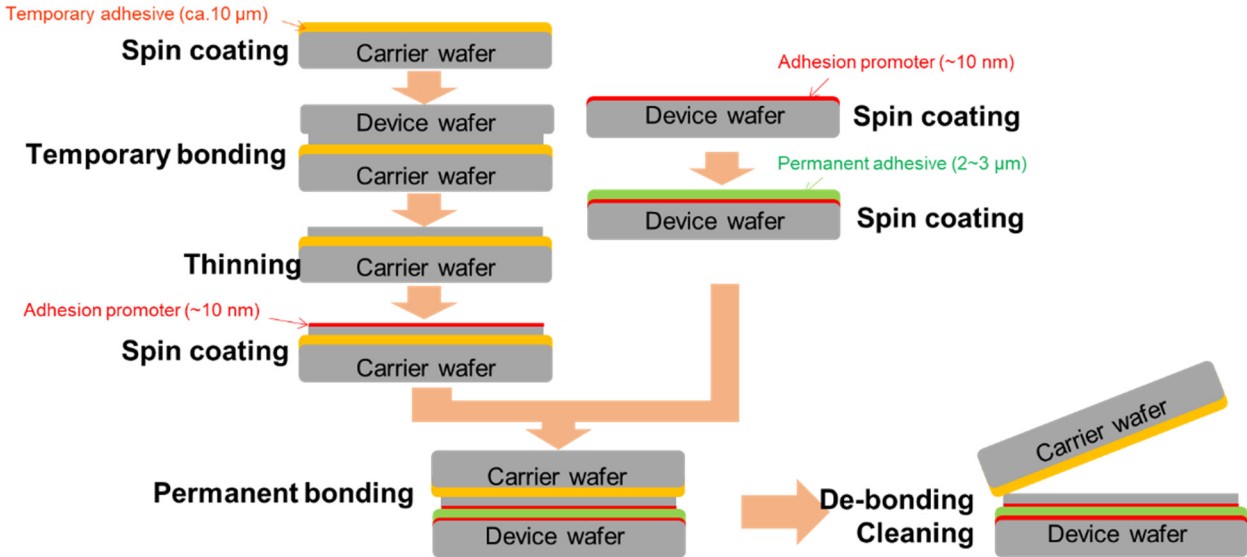

**Figure 10.** Process flow to evaluate temporary and permanent adhesives for design of experiment (DOE) of wafer stacking.

Permanent adhesives of DPAS100 have high thermal stability and have a maximum operating temperature up to 300 °C suited to the thermal budget in Back-End processes. Because the permanent adhesive needs to be cured within the operating temperature range of the temporary adhesive, its functional chemical groups and curing temperature are optimized. In order to increase the bonding strength between the Si surface and the organic polymer, a different chemical reactivity needs to be created between the two incompatible materials using an adhesion promoter. In this process, a silane coupling agent having an epoxy group was applied as an adhesion promoter. For the permanent adhesive layer, no outgassing by the Gas Chromatography-Mass Spectrometry (GC-MS) spectrum was observed during heating. The weight loss of the permanent adhesive after heating at 300 °C for 30 min was less than 1% by weight. These observations indicated that the adhesive layer had very little residual solvent, unreacted material, and degraded material. In addition, no delamination of the singulated thin wafer and no change in the structure of the permanent adhesive layer was observed after the Temperature Cycle Test (TCT) from −55 °C to 150 °C for 13.5 min hold time of 1000 cycles. The properties and process conditions for both the temporary adhesive and permanent adhesive are shown in Table 1.

**Table 1.** Properties and process conditions.

|  | **Permanent Adhesive on Adhesion Promoter** | **Adhesion Promoter** | **Temporary Adhesive** |
|---|---|---|---|
| Thickness | 0.5–5 μm | ~10 nm | 2–20 μm |
| Softening Temperature | around 50 °C | - | over 100 °C |
| Solidification | 135 ± 5 °C, 30 min and 170–195 °C, 30 min | 100–120 °C, 5 min | - |
| De-bonding | - | - | mechanical peeling at 80 kPa or over 200 °C |
| Cleaning Solvent | - | - | DTB-Cleaner |
| Modified Tape Peel Test of Stacked Wafers, After TCT 1000 Cycles | no delamination | - | - |

### 4.3. Through-Silicon-Via (TSV) Module

For bumpless TSV interconnects including RDLs, the Damascene method, a mature method based on a Cu/Low-k BEOL process [64] was used to simplify the processes. In the case of TSV processing, dry etching through the dielectrics in BEOL, device layer including shallow trench isolation (STI), Si, and adhesive layer was carried out. Bumpless TSVs with a small aspect ratio, for example <3, have the advantage of shortening the process time for both etching and metal filling compared with conventional deep TSVs. For instance, assuming that the etching rate follows the mass transport limit reaction, the etching times, t and $t_1$, at different TSV diameters, D and $D_1$, and depths, d and $d_1$, followed $t_1/t = (D_1/D)^2 \times (d_1/d)$; that is, $t_1/t = 0.1$ at $D = D_1$, d = 50 μm, and $d_1$ = 5 μm, which suggests 1/10th the etching time for the same TSV diameter and 1/10th the depth.

After TSV etching and wet cleaning, a low-temperature Plasma Enhanced Chemical Vapor Deposition (PECVD)-SiN or $SiO_2$ film was deposited to provide electrical insulation from the Si substrate. The barrier dielectric at the bottom of the TSV was removed by bias sputtering of Ar ions, and Ti/TiN (or Ta/TaN) and Cu were deposited on the barrier metal and the seed layer, respectively, by sputtering. For Cu plug interconnects and RDLs, Electrochemical Deposited Cu (ECD-Cu) was used. ECD-Cu planarization was carried out by chemical mechanical polishing (CMP) to polish-off of the Cu overburden.

Figure 11 shows the leakage current of TSVs varied with annealing temperature as a function of applied voltage, comparing Bosch and direct dry etching methods [65,66]. Since Bosch etching was carried out by repeated alternating isotropic-etching and deposition for sidewall passivation/protection, micro-steps called scalloping were formed in the side walls. The scalloping caused cracks and poor step coverage in the dielectrics and metal layers for thin films deposited by Chemical Vapor Deposition (CVD) and Physical Vapor Deposition (PVD). In contrast, anisotropic dry etching resulted in a smooth surface profile along the side wall and no discontinuous layer was observed. The leakage current in Bosch etching was one order of magnitude higher than that in anisotropic dry etching. The leakage current was caused by Cu diffusion at the side wall of the TSV, which took place at a thinner part of the dielectrics containing cracks. Thus, anisotropic etching is suitable for TSV interconnects and enables the use of low-aspect-ratio vias in the BBCube.

The stress inside the Cu was induced by a mismatch in the CTE between Cu and Si decreases with decreasing aspect ratio of the TSV. Figure 12 shows the results of the Finite Element Method (FEM) analysis of the maximum principal stress for different via heights with 10 μm thick device layers and BEOL interconnects [67]. The stress inside the Cu via with 110-μm thick Si was about twice that of the thinner, with a 30 μm thick via with a constant diameter. The Cu stress at the high-aspect-ratio via exceeded the yield stress (286 MPa) of Cu. The stress distribution showed the following critical points: (a) the BEOL region on the via side, (b) the interior of the via, (c) the bulk region under the via, and (d) the BEOL region under the via. The effect of Si thickness, TSV diameter, adhesive layer thickness and CTE of the adhesive material on the TSV stress was analyzed by sensitivity analysis of the DOE (Design of Experiments) method. The TSV compresses the device surface because the CTE of the adhesive (~50 ppm) was higher than those of Cu (16.6 ppm) and Si (2.6 ppm), and tensile stress was generated due to the strain around the BEOL. Stress at the center of the Cu plug decreased in proportion to the thickness of the Si wafer where the concentrated stresses at thicknesses of 20 μm and 100 μm were 225 MPa and 525 MPa, respectively. Thus, the small aspect ratio provided by an ultrathin wafer had the advantage of reducing stresses generated in the silicon itself, in the bottom and top Cu-TSVs, and in interface regions, even with different CTEs.

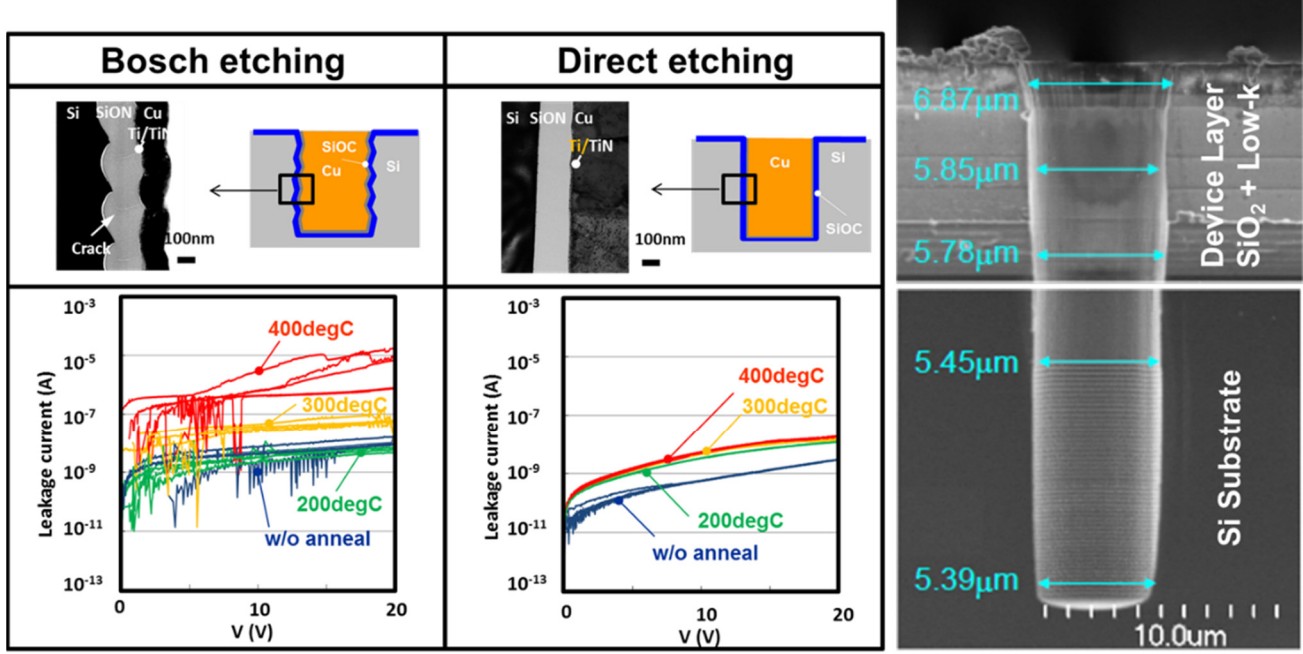

**Figure 11.** Schematic diagram, cross-sectional TEM images, and leakage current of two types of TSV samples made by Bosch etching and anisotropic dry etching (**left**). Cracks are observed in the Bosch-etched sample, which had a rough interface due to scalloping. The leakage current as a function of applied voltage after annealing at temperatures up to 400 °C was measured. With increasing temperature, the leakage current increased but was two orders of magnitude higher in Bosch etching. SEM images of TSV etched off through Cu/Low-k BEOL layer, device layer, and Si after optimization of scalloping shape (**right**). Fine etching profile through BEOL and Si is achieved.

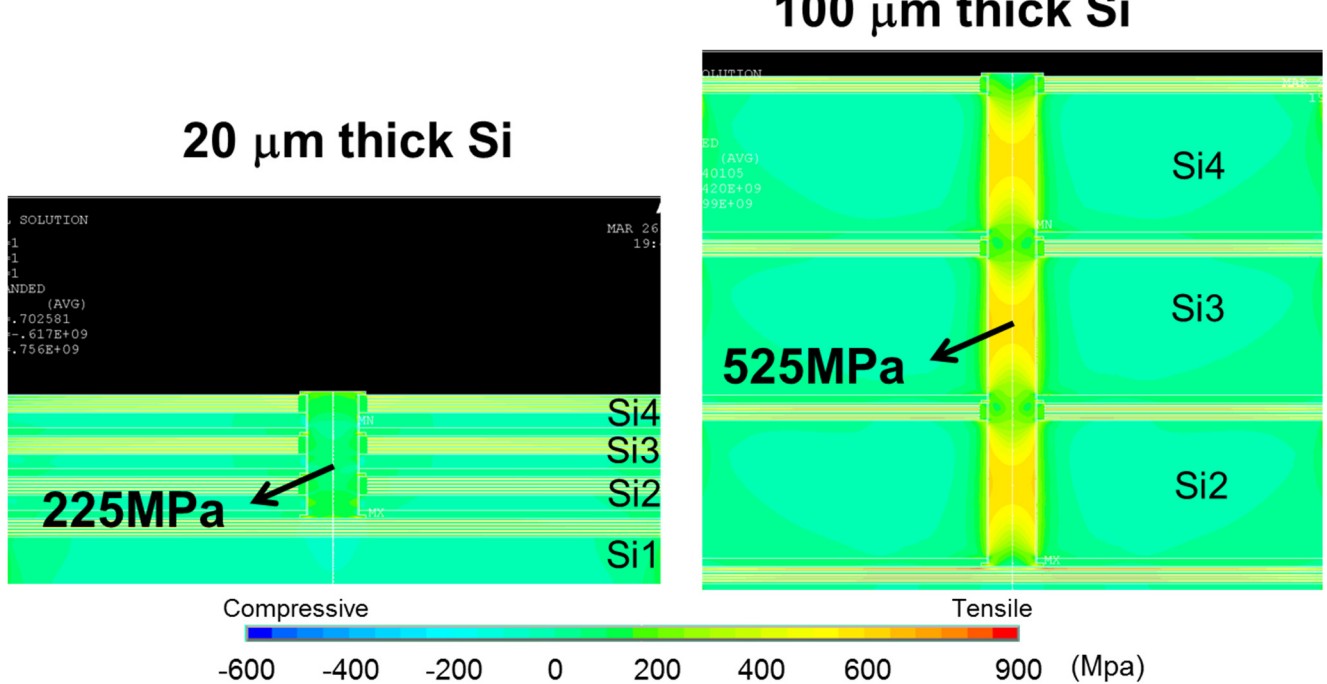

**Figure 12.** Stress simulation of Cu TSV using the FEM for a Si thicknesses of 100 μm (**right**) and 20 μm (**left**) after three wafers stacking. TSV diameter is 30 μm. A 10 μm Cu/low-k BEOL layer is formed on every wafer surface, and thus the depths of the TSVs are 110 μm and 30 μm, respectively.

### 4.4. Singulation/Packaging Module and Reliability

After multi-level wafer stacking and TSV interconnects, when the stacked die was applied to the interposer, the same procedure as in conventional packaging (micro-bumps, singulation by dicing, die attach) was followed. After dicing the seven-level wafer stack, the adhesive layer and silicon chips were found to be free of defects or delamination. After the stacked chips were packaged with epoxy resin, they were subjected to heat stress testing at temperatures of −65 °C to 150 °C. Scanning acoustic tomography (SAT) is adopted for internal observation, and after up to 100 repeated heat stress tests, no delamination and voids were found at the interfaces between the molding compound and chips, nor at the chip stack interfaces [68].

A temperature cycling test (TCT) following JESD22A-104 was performed to examine whether the bumpless structure is able to withstand the mechanical stresses caused by extreme temperature variation. The daisy chain (total number of vias $n$ = 216) resistance of the structure with via bottom cleaning showed negligible change after the TCTs, indicating that the structure can tolerate extreme temperature changes despite the presence of a polymer with a high CTE within the structure, as shown in Figure 13a [69]. In general, a high moisture content may decrease the glass transition temperature of the polymer and damage the structure. A high accelerated stress test (HAST) following JESD22A-118 was performed to further investigate the reliability of the structure under specified temperature and moisture conditions. In Figure 13b, the daisy chain ($n$ = 216) resistance of the structure with via bottom cleaning only slightly increased after the HAST, showing that the bumpless structure can successfully protect the polymer from moisture. According to the TCT and HAST tests, the designed structure has good fabrication integrity and is highly reliable.

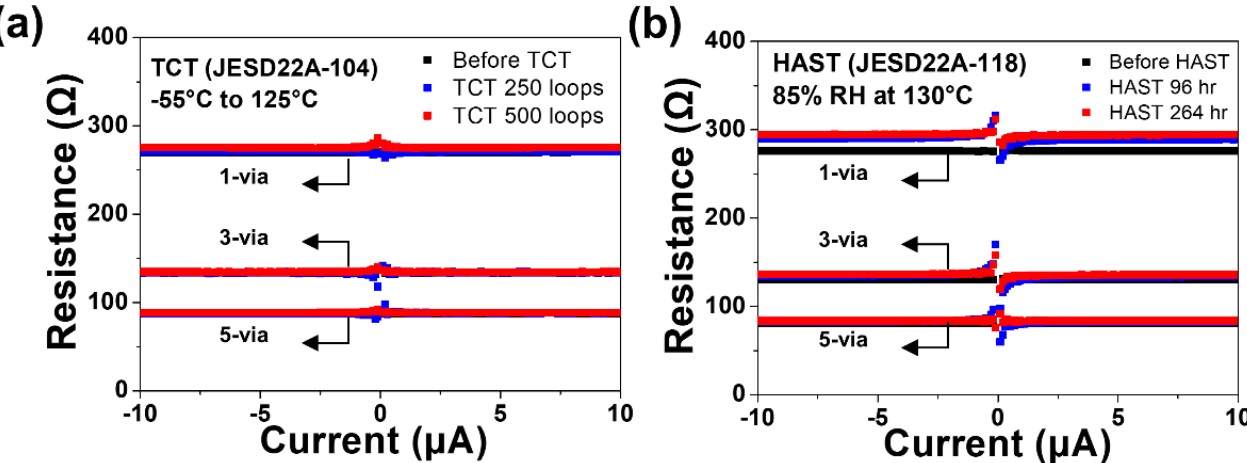

**Figure 13.** Rsistance change of daisy chain ($n$ = 216) measurement before and after (**a**) thermal cycling test (TCT) and (**b**) highly accelerated stress test (HAST), respectively.

Electromigration (EM) test and SEM analysis was performed to determine the failure site, as shown in Figure 14. The cross-sectional images of bumpless TSVs before and after current stressing indicate that the failure site is located at the bottom RDL close to the corner with the TSV, which is consistent with the simulation results. The failure at this location mainly results from the small thickness of the RDL. Thus, the mean time to failure (MTF) of the bumpless structure can be increased by increasing the RDL thickness [70]. A comparison of the electromigration characteristics between the bumpless TSV structure and the conventional TSV structure with microbumps is shown in the table in Figure 14, indicating that the bumpless TSV technology has better mechanical properties and the ability to withstand longer current stressing time [71].

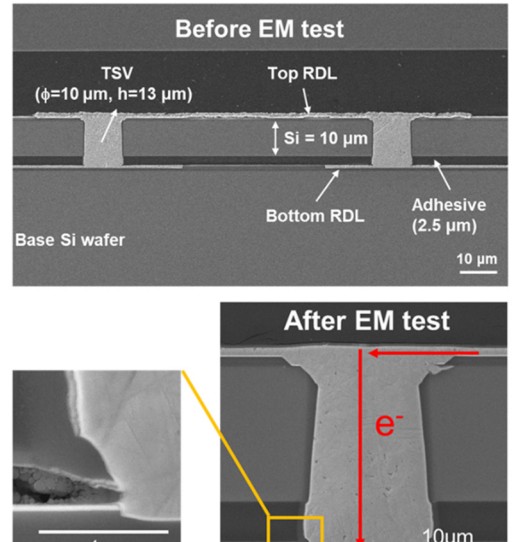

| Ref. | Akamatsu et al. ECTC 2016 | This work |
|---|---|---|
| **Structure** | Conventional TSV structure with microbump | Bumpless TSV structure |
| **Diameter** | TSV (10 µm) IMC joint (30 µm) | TSV (10 µm) |
| **Temperature** | 100 °C | 200 °C |
| **Current density** | $3.8 \times 10^5$ A/cm$^2$ | $7 \times 10^5$ A/cm$^2$ |
| **Criteria** | 10% | 10% |
| **MTF** | 4 hr | 137 hr |

**Figure 14.** Cross-sectional SEM images of bumpless TSV interconnects before and after electromigration test (EM) (**left**). A comparison of electromigration characteristics between bumpless TSV structure and the conventional TSV with microbump (**right**) [71].

## 5. BBCube COW Processes

### 5.1. Heterogenous 3D Integration Process

For heterogeneous integration using chiplet logic devices, memory, and passive devices such capacitors, BBCube COW was developed [72]. Figure 15 shows the bumpless COW process flow used for a 3D functional interposer. First, a permanent adhesive material of Bis-benzo-cyclobutene (DOW, CYCLOTENE™ 3022-46) was coated on a 300 mm Si wafer (base wafer) to a thickness of 5 µm. The Si base wafer was patterned to make fiducial marks on the surface to determine the Si capacitor die placement positions. A dummy Si for reducing the volume ratio of dies and mold material was bonded to the Si base wafer. Then. a Si capacitor die was placed on the adhesive from the front side with a surface mounter tool. The Si capacitor was a commercial product (Murata Manufacturing Co., Ltd., Kyoto City, Japan, EMSC series), with dimensions L = 3.07 mm, W = 2.07 mm, T = 100 µm, a capacitance of 1 µF and an equivalent series resistance (ESR) of 100 mΩ. Si capacitors based on deep-trench metal-oxide-semiconductor (MOS) capacitor technology combined with a unique mosaic design and distributed trench capacitors were developed, as shown in Figure 16. As an example, a 100 nF equivalent series inductance (ESL) Si capacitor was made of 200 elementary cells of 470 pF distributed over the chip and combined in parallel with a 10 pF metal insulator metal (MIM) capacitor to lower the impedance at higher frequencies. A Si capacitor is one candidate for overcoming the scaling issue faced by capacitor components.

The permanent adhesive was cured to bond the Si capacitor and dummy Si after die attachment. Epoxy resin with silica-based filler was molded on the die-attached side of the base wafer with a compression molding method. Then, epoxy resin was thinned to a thickness of several tens of micrometers above the Si capacitor. To reduce the wafer warpage in the COW process, a 300 mm Si wafer (carrier wafer) was bonded on the side of the thinned resin mold. The base wafer was thinned down from a thickness of 775 µm to 20 µm with grinding and polishing. The TSV and re-distribution line (RDL) were formed by the Damascene method to make interconnects between the Si capacitor and the RDL.

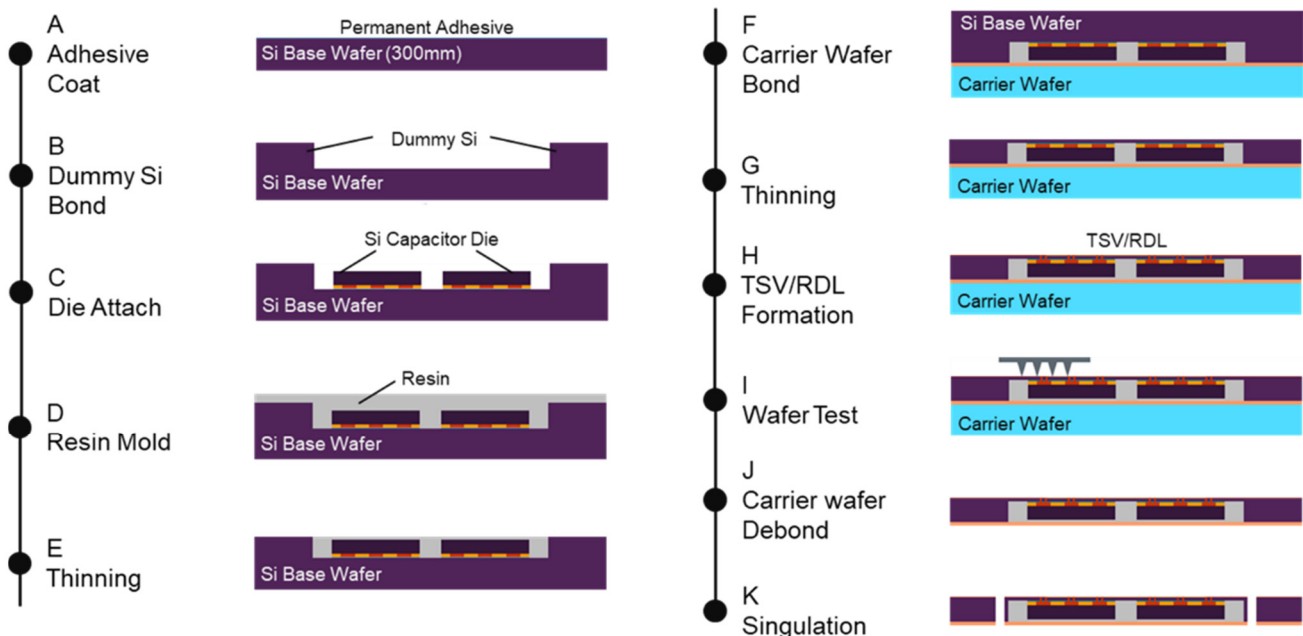

**Figure 15.** Bumpless COW process flow. Face-down die attachment and Front-side bumpless TSV interconnects are carried out.

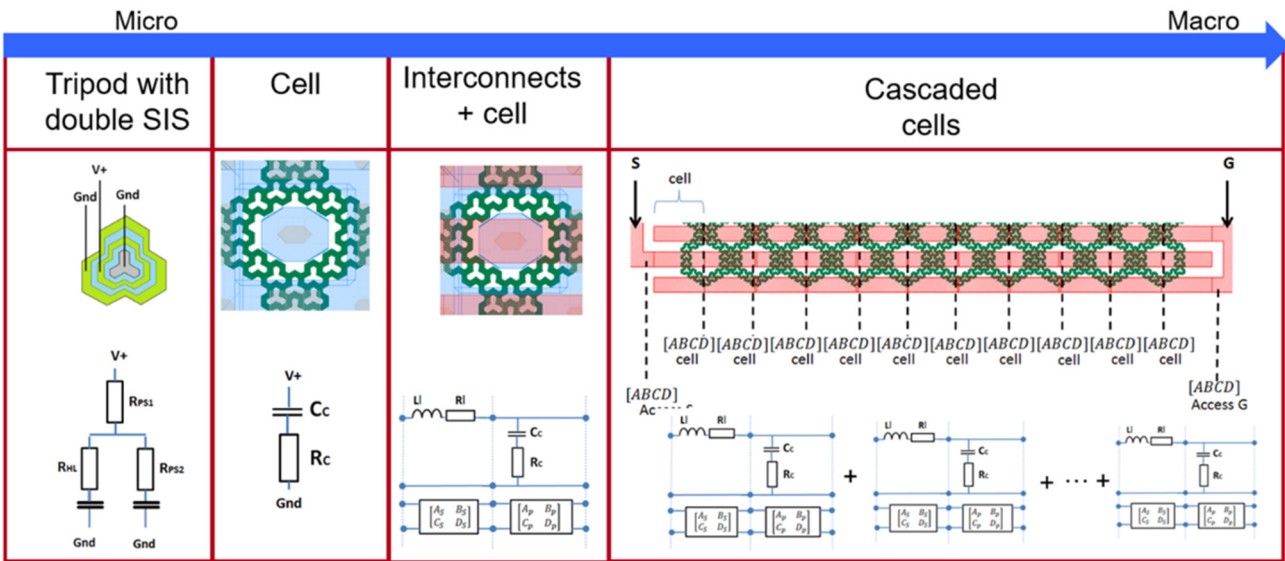

**Figure 16.** Design of Si capacitor based on deep-trench metal-oxide-semiconductor (MOS) capacitor technology combined with a unique mosaic design and distributed trench capacitors. Die size is L = 3.07 mm, W = 2.07 mm, T = 100 μm. A capacitance and ESR are 1 μF and 100 mΩ, respectively.

### 5.2. Wafer Warpage Control

Wafer warpage, due to a mismatch in the CTE between organic materials and Si, is a major problem in wafer-level-packaging (WLP) integration. Huge wafer warpage causes wafer cracking and even wafer breakage in the worst case. Even small wafer warpage in the millimeter range causes wafer chucking problems in tools such as the grinder/polisher and wafer bonder, and all vacuum process tools for the TSV/RDL processes. To satisfy the vacuum process and the TSV/RDL formation step, wafer warpage should be less than 300 μm, even though the wafer has a multi-layer structure that includes Si capacitors and molded resin. In particular, the molded resin has a quite different CTE compared to Si, so

how to reduce the volume of molded resin is the most important parameter for controlling wafer warpage in the COW process [73].

Figure 17 shows wafer warpage as a function of the molded resin thickness (a), and the definition of the mold cap and its thickness (b), wafer warpage increases linearly as the mold resin thickness increases. Wafer warpage became more than 1.6 mm at the resin thickness of 300 μm, indicating a smile shape. The mold cap was set to 100 μm for bumpless COW process integration. Figure 18 shows the details of wafer warpage in each step of the COW process. In the resin molding step, wafer warpage shows a maximum value of 800 μm. After mold thinning, wafer warpage decreased significantly from 800 μm to 100 μm due to the reduced volume of the molded resin. At the beginning of the TSV/RDL step, wafer warpage was 100 μm, which is low enough to process the TSV/RDL formation step as a result of an appropriate mold cap.

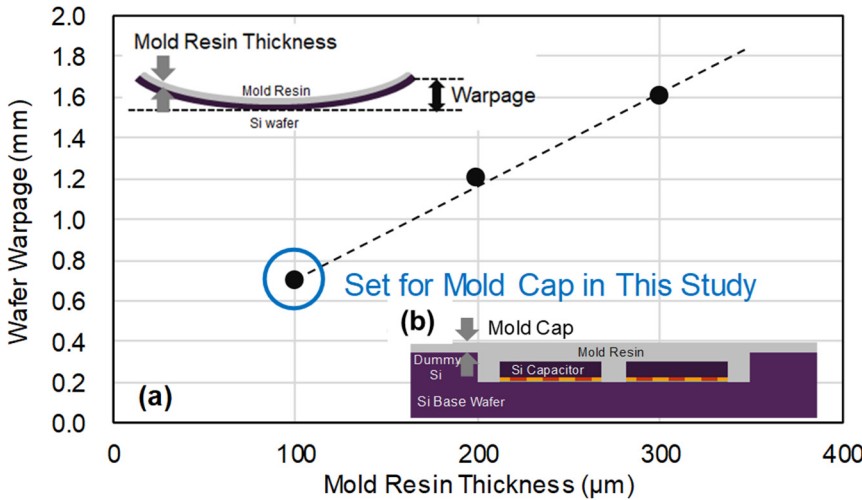

**Figure 17.** Wafer warpage in resin molding step. (**a**) Wafer warpage as a function of molded resin thickness, and (**b**) cross-sectional image of mold cap in COW process.

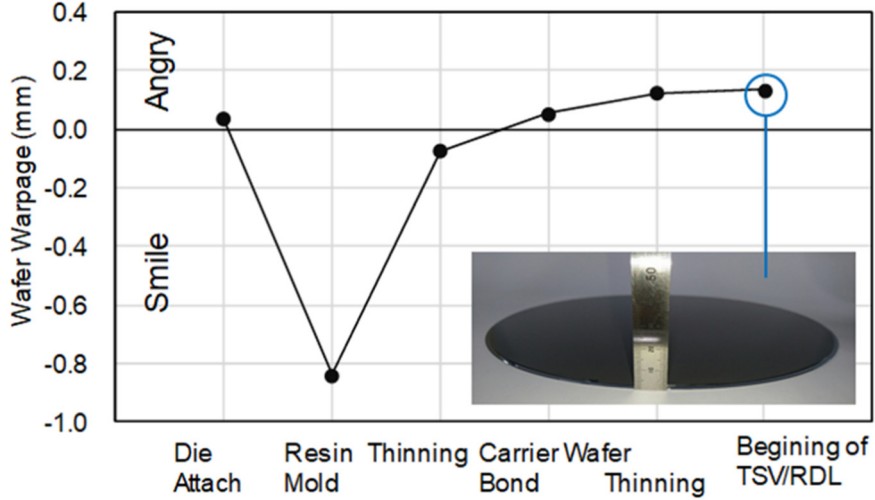

**Figure 18.** Details of wafer warpage in each step of COW process.

### 5.3. Accuracy of Die Placement and Reduction of Voids in Adhesive

Die placement is one of the primary technologies in the COW process, whether COW has bumps or not. This is because the die placement accuracy strongly affects the reliability of connectivity and the allocation of high-dense TSVs [74]. In addition, to increase the placement accuracy, there is a trade-off in the throughput. For the bumpless COW with adhesive, misalignment of the Si capacitor in the die placement tool directly causes

TSV interconnect failure. Even if the die placement tool has highly accurate placement positioning of the Si capacitor, die shifting may occur in the adhesive curing step because the adhesive sometimes contains voids that move to the outer region of die. In addition, when the large voids are located at the TSV region, discontinuous barrier layer formation of CVD $SiO_2$ and PVD metals occurs, which causes void formation in the ECD-Cu metallization and leakage current between TSVs.

To overcome voids and die shifting, a cyclical pressure profile from low-pressure to atmospheric pressure of a pressure oven was used for the adhesive curing step. Figure 19 shows the observation of voids in adhesive and the die placement accuracy before and after the adhesive curing. Figure 20 shows a cross-sectional SEM image of the Si capacitor die after the die attaching step and the adhesive curing. No voids are observed after the cyclical pressure, and die shifting is maintained within $+/-25$ μm even after the adhesive curing. As a result of the low number of voids and the small die shifting, the thickness of the adhesive between the Si base wafer and the Si capacitor pad was stabilized at 5.2 μm.

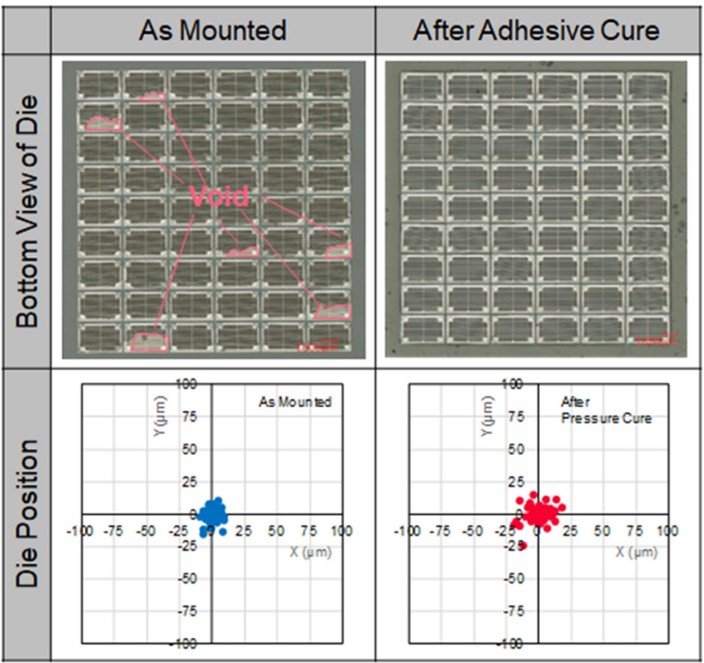

**Figure 19.** Observation of void in adhesive between the base wafer and Si capacitor (pictures in **top-left** and **right**); and die placement accuracy before and after the adhesive curing step (**bottom-left** and **right**).

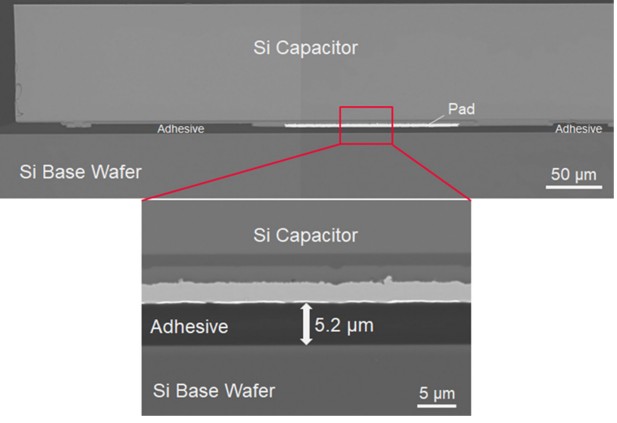

**Figure 20.** Cross-sectional SEM image of Si capacitor die after die attaching step. No voids at the pad interface are observed.

### 5.4. Process Sequence of TSV/RDL

The process sequence of TSV/RDL formation is shown in Figure 21. TSV/RDL formation is based on Cu Dual Damascene interconnects. After Si capacitor placement by the COW process as described, the Si base wafer was thinned down to 20 μm with a grinding and polishing tool (DISCO, DGP8761). Wafer thinning is usually carried out in three steps: (a) coarse grinding for a high removal rate of Si, (b) fine grinding for reducing back-side damage, and (c) stress relief such a CMP for removal of damage. According to measurements of the thickness uniformity of the thinned Si, a very low total thickness variation (TTV) of less than 2.0 μm, <10% un-uniformity at 20 μm of remained Si was achieved within the 300 mm wafer, which was low enough for the subsequent steps. After thinning of the Si base wafer, a dielectric layer such a $SiO_2$ was deposited using a low-temperature plasma-enhanced CVD. Silicon dioxide was patterned for the RDL using the lithography and an etching process. TSV formation is carried out with lithography and etching of the Si and adhesive until the pad of Si capacitor. To protect the Si sidewall of the TSV from Cu, a $SiO_2$ liner was deposited conformally and then etched to remove the bottom $SiO_2$. Finally, Cu metallization and planarization were performed. These steps were characterized as a *TSV-last* process from the front side. Via bottom cleaning, such as with $O_2$ plasma, wet cleaning, and Ar sputtering, removed contaminants, including residues and by-products from the etching process.

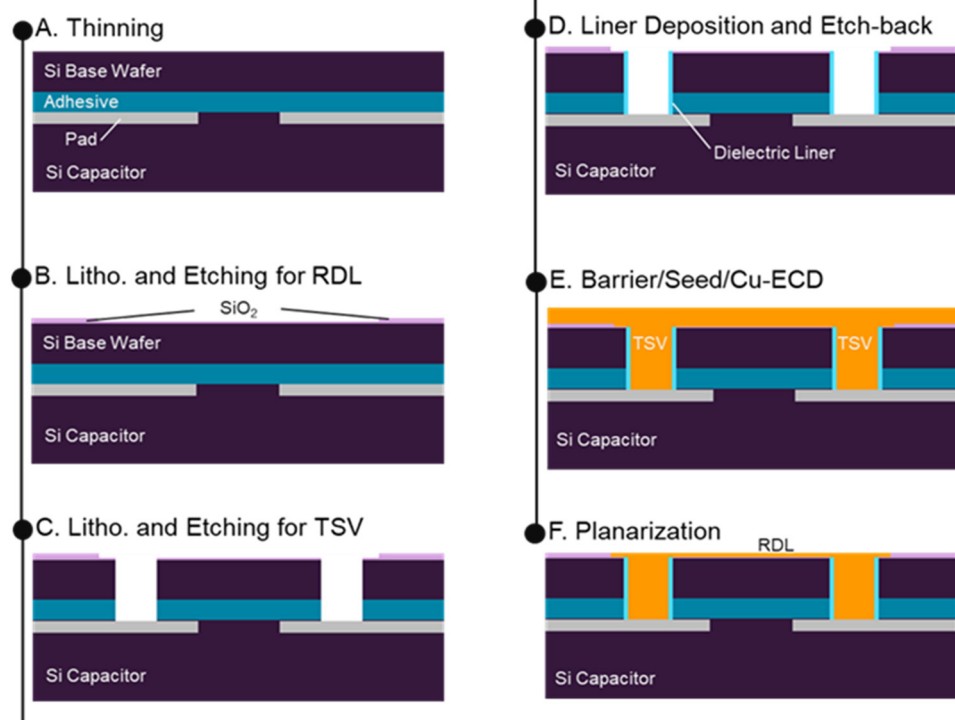

**Figure 21.** Process sequence of TSV/RDL formation after chip placement and wafer bonding.

The length of the TSV interconnect was 25 μm, which is equal to the total thickness of the thinned Si and the adhesive layer, and the diameter was 10 μm, as shown in Figure 22. A fine plug profile without voids was observed in a cross-sectional SEM image of the TSV/RDL. There was no significant oxide residue in the interface between the TSV and Si capacitor pad. The TSV resistance measured with the Kelvin method was 10 mΩ and had excellent uniformity, which indicated low local warpage of the Si capacitor and low curing shrinkage of the adhesive embedded in the interposer. If the interposer had huge warpage and shrinkage, the resistance of the TSV would have a large deviation and open failure due to unstable contact resistance.

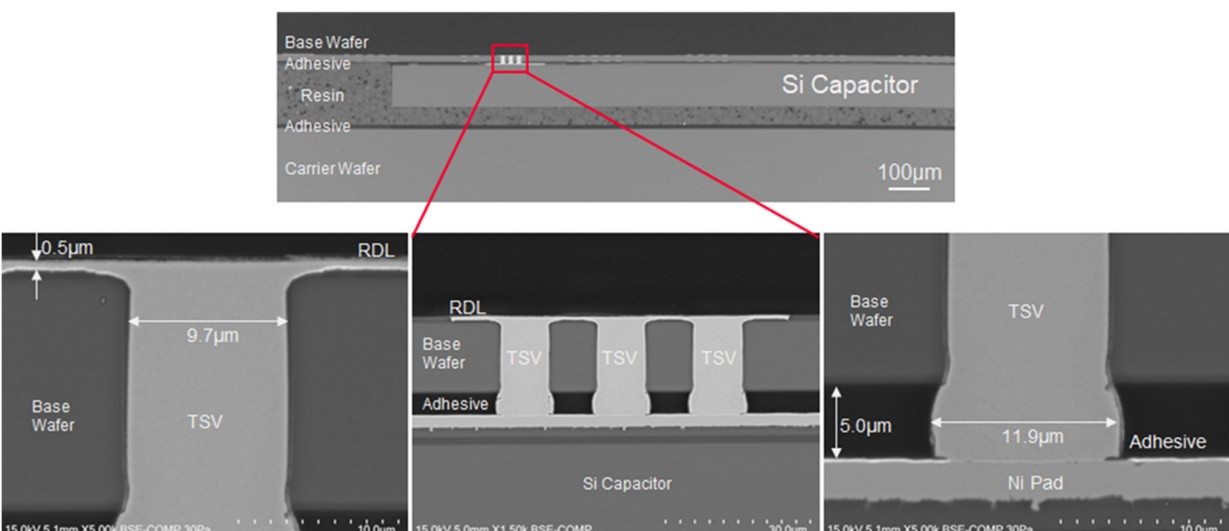

**Figure 22.** Cross-sectional SEM images of TSV/RDL. Fine plug profile without voids and no significant oxide residue in the interface between the TSV and Si capacitor pad are observed.

*5.5. Electrical Characteristics of Embedded-Si Capacitor in COW*

The electrical characteristics for high frequency of the Si capacitor embedded in the functional interposer were evaluated by measuring the S-parameter with the shunt through method from 10 kHz to 8.5 GHz. For the measurement, an Al contact pad is formed on the RDL. A vector network analyzer (VNA) was used for the measurement. The impedance, Z, was calculated with Equation (1), shown below. Then, the capacitance, C, and ESR were calculated with Equation (2). Here, $Z_0$ is a reference impedance of 50 Ω, $S_{12}$ is the measured S-parameter, and $\omega$ is angular frequency.

$$Z = \frac{Z_0}{2} \times \frac{S_{12}}{1 - S_{12}} \tag{1}$$

$$Z = ESR + \frac{1}{j\omega C} \tag{2}$$

The measured and calculated RF characteristics are shown in Figure 23, where (a) is the impedance, (b) is the capacitance and (c) is the ESR. The measured capacitance is 0.95 μF at 100 kHz, and ESR is 43.5 mΩ at 100 MHz. These results for the embedded Si capacitor are same as the values measured on the wafer of Si capacitor before the COW process, which means that the bumpless COW process did not have any electrical loss.

*5.6. Benefits and Performance of Bumpless Functional Interposer*

From the point of view of electrical performance using bumpless COW integration, the principal advantages of a 3D functional interposer are its significantly low energy consumption and low parasitic capacitance, which were expected due to the shortest interconnect length between MPUs and the Si capacitor. However, the TSV formed a metal-oxide-semiconductor (MOS) capacitor at the sidewall of TVS interconnects as well as a Si capacitor, so the parasitic capacitance of the TSV should be considered. Parasitic capacitance of the TSV consists of accumulation capacitance and depletion capacitance along the interconnect line. In the case where the maximum accumulation capacitance is considered, the accumulation capacitance was the same as the liner oxide capacitance, $C_{ox}$, as expressed by previous studies [75]:

$$C_{ox} = \frac{2\pi\varepsilon_0\varepsilon_{ox}l_{TSV}}{\ln\left(\frac{r_{TSV}+t_{ox}}{r_{TSV}}\right)} \tag{3}$$

where $\varepsilon_0$ is the permittivity of vacuum ($8.854 \times 10^{-12}$ F/m), $\varepsilon_{ox}$ is the relative permittivity of the liner oxide (in this study, the liner oxide is $SiO_2$, hence $\varepsilon_{ox}$ is 3.8), $l_{TSV}$ is the length of the TSV, $r_{TSV}$ is the radius of the TSV Cu, and $t_{ox}$ is the thickness of the liner oxide.

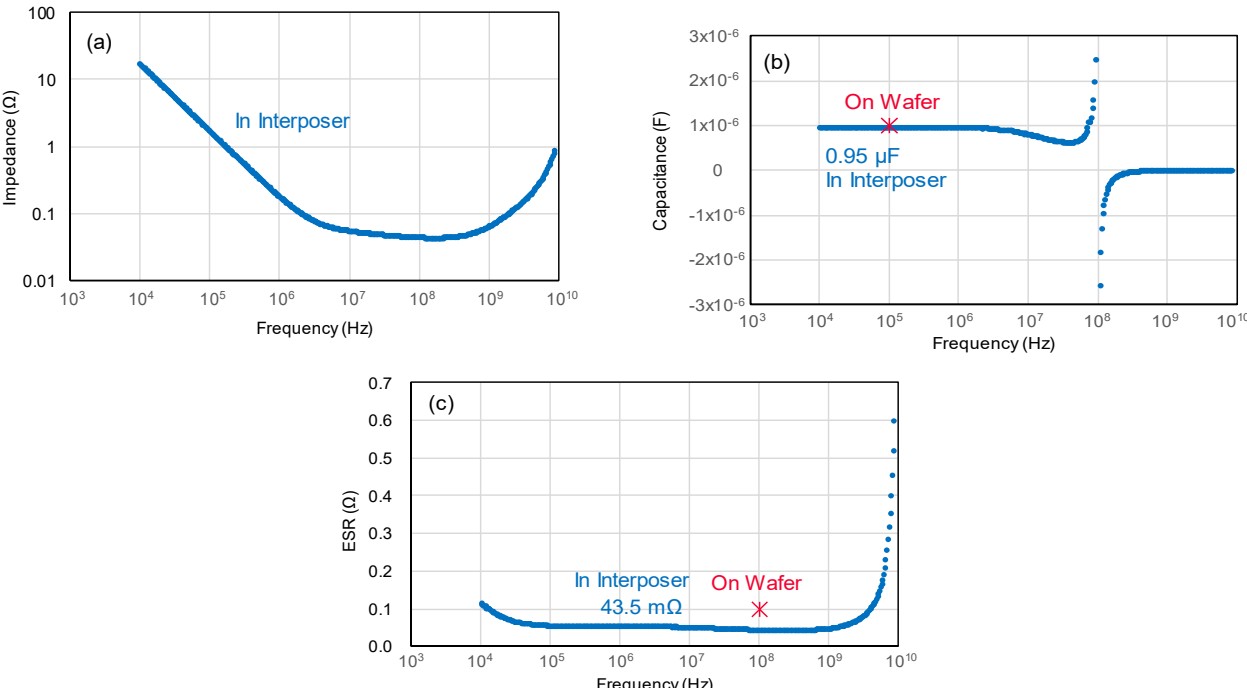

**Figure 23.** RF characteristics of Si capacitor embedded in functional interposer from 10 kHz to 8.5 GHz: (**a**) impedance, (**b**) capacitance, (**c**) ESR, compared to the Si capacitor before COW process indicated "On Wafer" red symbol.

Figure 24 shows the schematic diagram of TSV structure used to estimate parasitic capacitance, and the calculated parasitic capacitance as a function of the TSV or line length. Parasitic capacitance of the TSV used in our 3D functional interposer is calculated by Equation (3). The parasitic capacitance was significantly reduced to 1/150th compared to that of 2.5D, side-by-side capacitor layout. This extreme reduction in the parasitic capacitance was able to realise lower noise of the power supply for MPUs, and thus a lower applied voltage $V_{dd}$ such a <0.7 V could be used, taking the power distribution network (PDN) into account. As a result of the lower $V_{dd}$, the power consumption was decreased as the power consumption is proportional to $V_{dd}^2$.

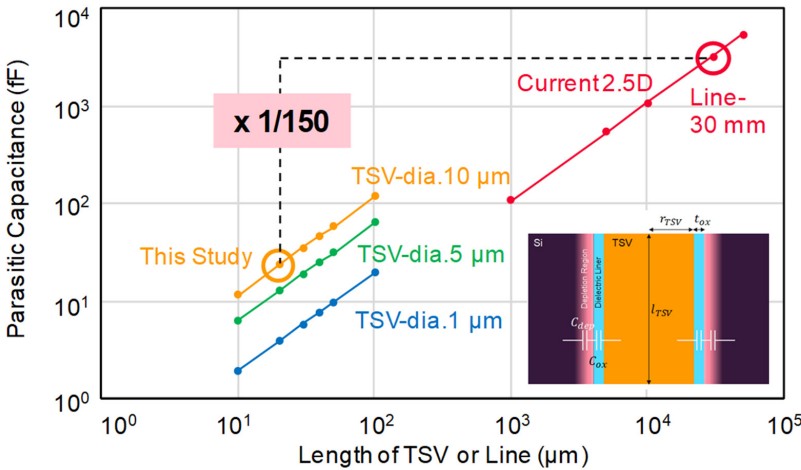

**Figure 24.** Parasitic capacitance as a function of the length of TSV or wiring line.

## 6. BBCube Technology Roadmap

Since the bonding process after thinning with a support wafer allowed thinning of silicon wafers down to 4 μm without any degradation of the device characteristics, the total wafer thickness, including the device layer and the adhesive layer was only 10 to 20 μm. This is 1/3rd to 1/5th the thickness of conventional bump interconnects using TSVs. Therefore, even if the number of stacked wafers is 100, assuming that the wafer thickness is 10 μm, the total thickness after stacking is 1 mm. This total height satisfies current packaging standards. Following these multilevel stacking processes, when four, eight, sixteen, etc. of these devices are stacked with a conventional memory device fabricated by a memory density of 30 Gb/cm$^2$, e.g., 22 nm technology, the total capacity of the 3D memory device can be linearly increased to 120 Gb, 240 Gb, 480 Gb, etc., respectively, as shown in Figure 25.

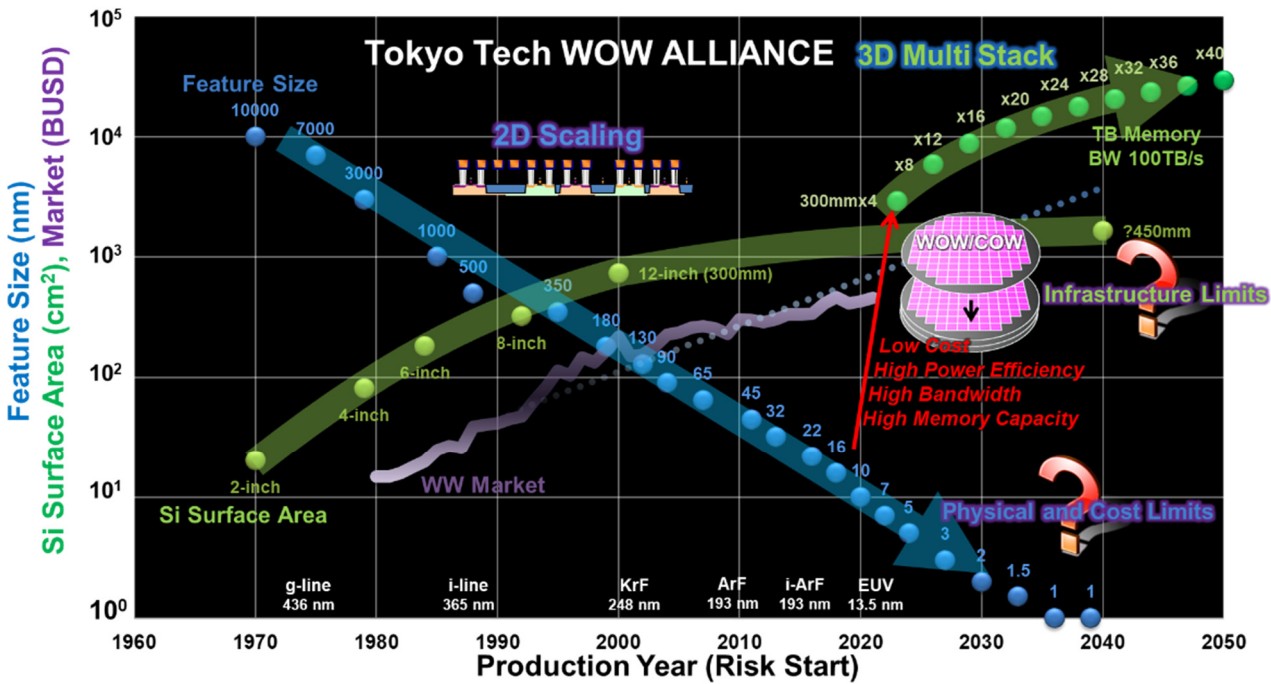

**Figure 25.** Historical trends and prospect of feature size of transistors, Si wafer area, and semiconductor market. See for example; Available online: http://www.wow.pi.titech.ac.jp/index.html (accessed on 25 December 2021).

Terabit-capacity 3D memory can be realized by stacking 40 layers. In contrast, to achieve equivalent capacity with a single wafer using extreme scaling would require 1 nm node technology, e.g., equivalent dimension about four times of the Si–Si bond length $d_{Si–Si}$ of 0.23 nm. Consequently, innovative technology not only for 3D transistors but also for 3D chip stacks is needed, as described in Section 2. Considering the technology roadmap, the issues of scaling technology and technology for fabricating 3D structures are often discussed separately. It has been considered that the packaging may take charge of 3D structure. However, these two technologies are not always mutually exclusive. Scaling would be relieved of the stringent requirements by using 3D high-density integration technology combined with mass-production technology. In other words, a sufficiently long learning period would be ensured, and further cost reductions could be expected by concentrating on the control of variations among generations and shortening the process.

Figure 26 shows a schematic diagram of the chip-level configuration for die-to-die connections. The configuration is an evolution from side-by-side to chip-stack in order to reduce signal latency, IR drop, and footprint on the package board. The BBCube is one candidate that satisfies those requirements. The bumpless connections and ultra-thinning enable the shortest wiring and high density TSVs, as well as improved misalignment in the

wafer stacking. High-density TSVs are useful because the parallel communication provides high bandwidth. According to the above capabilities, the BBCube architecture provides a solution to the long-standing discussion regarding signal propagation, power distribution, and heat dissipation in the high dense LSIs [76–78], described in the following sections.

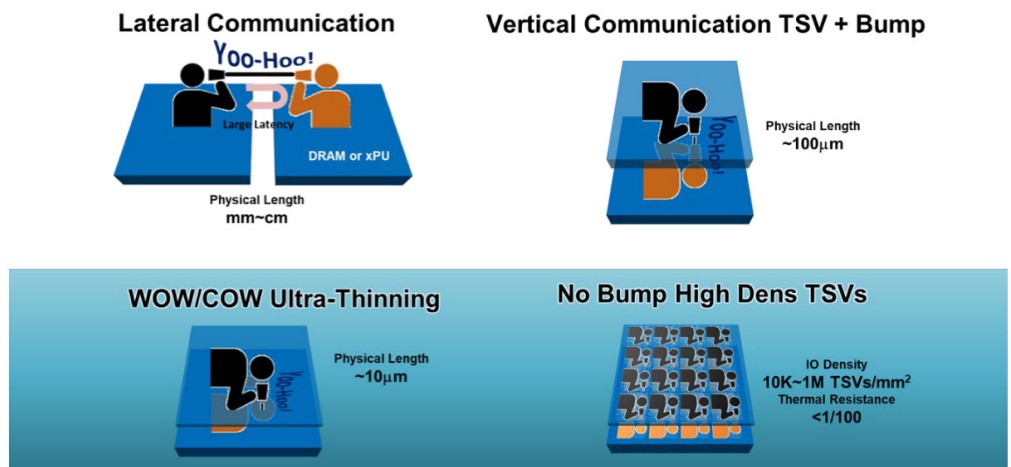

**Figure 26.** Schematic diagram of chip-level configuration. A physical length of side-by-side (lateral communication) and vertical stack is millimeters–centimeters and about 100 µm, respectively. With no bumps and ultra-thinned wafer, the physical length becomes approximately 10 µm and a high TSV density of more than $10^4/\text{mm}^2$ can be designed.

In fact, the bandwidth of recent high-bandwidth memory (HBM) tended to saturate due to bump pitch constraints, as shown in Figure 27. In the case of BBCube, an order of magnitude higher bandwidth can be realized, since BBCube uses high density TSVs and a novel memory architecture. According to the WOW Alliance, TSV pitch will be narrowed every three years, taking the maturity of bonding alignment into account.

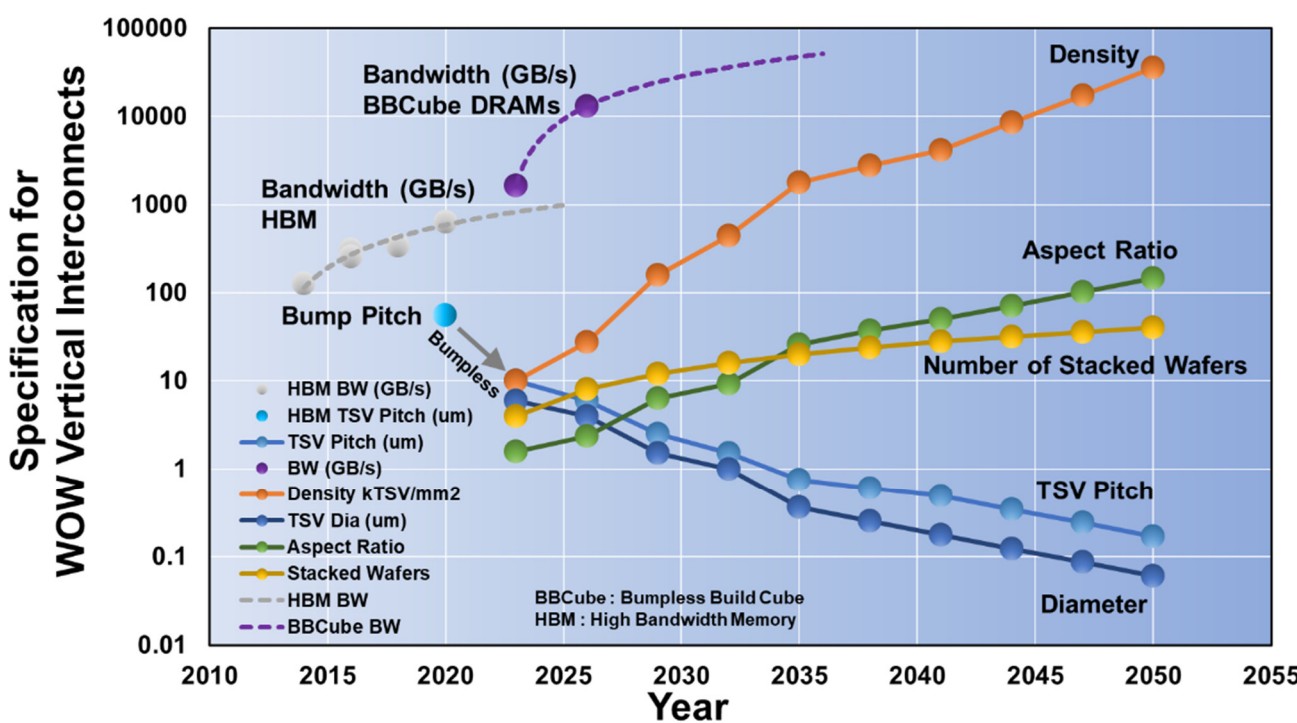

**Figure 27.** Trends of vertical interconnects of TSVs according to WOW Alliance.

## 7. BBCube Memory

There are three key challenges in the history of computing systems: (1) size reduction, (2) reduced power, and (3) higher speed. Among these key elements, size reduction is the most imperative challenge, because both low power and high speed can be achieved by size reduction itself. Figure 28 shows the computing system roadmap. In 2035, the target device volume will be 50 mm$^3$ with a power consumption of 0.5 mW, according to the extrapolating trend. Such a device might be something similar to an AI robotic bee, with CPU/GPU, DRAM [79], NAND flash memory [80,81], and sensors. It would serve human users, so that the AI robotic bee could observe the users' surroundings, protect them, and serve as an administrative secretary.

### Computing System

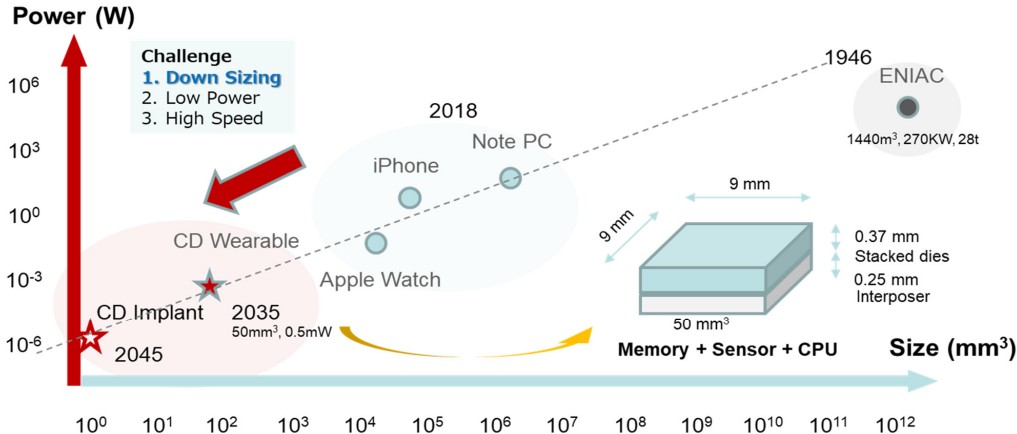

*Downsizing endows both low power and high speed!*

**Figure 28.** Computing system roadmap for the power consumtion as a function of system volume.

## 8. BBCube DRAM

TSVs with micro-bumps are conventionally used for the high-bandwidth memory (HBM) [82–87], as shown in Figure 29. However, there are several issues when using micro-bumps. One major problem is that it will be difficult for even HBM to catch up with the increasing speed of GPUs or CPUs. For example, Pascal manufactured by NVIDIA has a processing speed of 1 TB/s, so that four sets of HBMs with 256 GB/s would have to be used. GPU/CPU vendors are constantly striving to increase the speed of their products, for example, to 2 TB/s and 4 TB/s, focusing on AI systems. HBM will have to increase the I/O pin speed by 2.5 times, such as the 5.0 Gb/s/pin [87] from the 2.0 Gb/s/pin [84], and therefore, power and heat will also be increased.

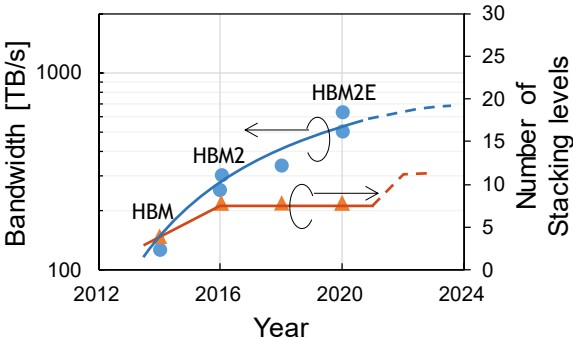

**Figure 29.** HBM road map with micro-bumps, where HBM [82], HBM2 [83–85], HBM2E [86,87], respectively.

### 8.1. Electrical Characteristics of BBCube

The electrical characteristics of the BBCube structure were calculated by 3D EM field analysis and compared to conventional 3D integration (3DI) with micro-bumps [34]. The TSV model used for conventional 3DI with micro-bumps, such as HBM, is shown in Figure 30a, and the TSV model used for BBCube is shown in Figure 30b. In conventional 3DI, on top of the TSVs, bumps consisting of copper pillars and solder were formed [88]. We assumed dimensions equivalent to those of HBM. The stacking pitch was 87 μm, and the TSV pitch was 55 μm. In comparison, BBCube's Si was thinned to 4 μm, and the stacking pitch became 10 μm. The TSV pitch was 12 μm. The physical dimensions and material properties are shown Tables 2 and 3 respectively.

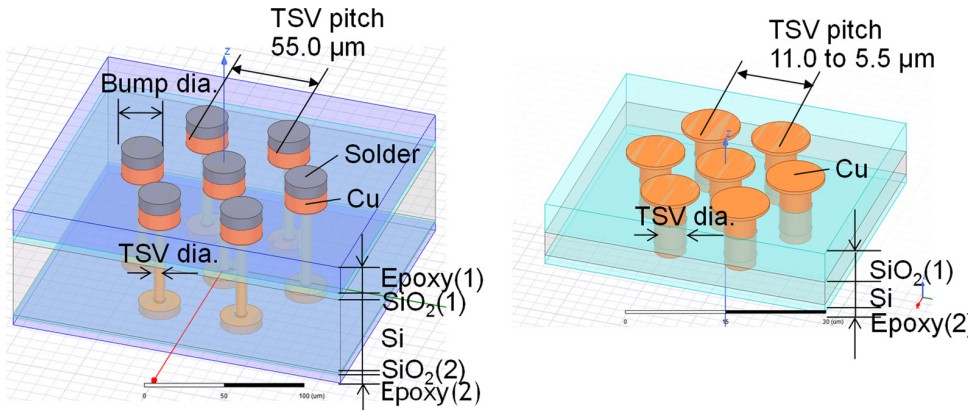

**Figure 30.** TSV model for 3D EM field analysis; (**a**) 3DI with micro-bumps and (**b**) BBCube.

**Table 2.** Physical dimension.

| | | 3DI w/Micro-Bumps | BBCube WOW | |
| --- | --- | --- | --- | --- |
| | | | **Gen1** | **Gen2** |
| | Epoxy (1) | 19.5 | - | |
| | Epoxy (2) | 6.0 | 1.5 | |
| | SiO$_2$ (1) | 5.0 | 4.5 | |
| Thick. (μm) | SiO$_2$ (2) | 1.5 | - | |
| | Si | 55.0 | 4.0 | |
| | Total | 87.0 | 10.0 | |
| TSV dia. (μm) | | 8.0 | 4.0 | 2.0 |
| TSV pitch (μm) | | 55.0 | 12.0 | 5.5 |

**Table 3.** Material properties.

| | Relative Permittivity | Bulk Conductivity (Siemens/m) |
| --- | --- | --- |
| Copper | 1.0 | $5.8 \times 10^7$ |
| Si | 11.9 | 10 |
| SiO$_2$ | 4.0 | 0 |
| Solder | 1.0 | $7 \times 10^6$ |
| Epoxy | 3.6 | 0 |

Figure 31a shows the frequency characteristic of the TSV capacitance. Due to a slow-wave mode [89], it increased below 3 GHz. As shown in Figure 31b, the liner thickness determines the TSV capacitance below 3 GHz. The TSV diameter and Si thickness also determines the TSV capacitance, which can be reduced by employing BBCube. As shown in Figure 31c, the TSV pitch did not affect the TSV capacitance. Therefore, when the TSV diameter was 5 μm, BBCube was able to shorten the TSV pitch to 11 μm without increasing the capacitance. Moreover, BBCube was able to shorten the TSV pitch to 5.5 μm when

the TSV diameter was 2 μm. Compared to the conventional 3DI, the TSV capacitance in the case of BBCube became 1/20th. The frequency dependence of the TSV resistance as shown in Figure 31d increased over 5 GHz due to skin effect, but this was higher than the operating frequency of BBCube, so it did not have any influence. The TSV inductance as shown in Figure 31d was flat to frequency. Due to the shorter TSV, the resistance and inductance in the BBCube case were much smaller than conventional 3DI. In the case of inductance, it reduced 1/10th to 1/15th than that of the conventional one.

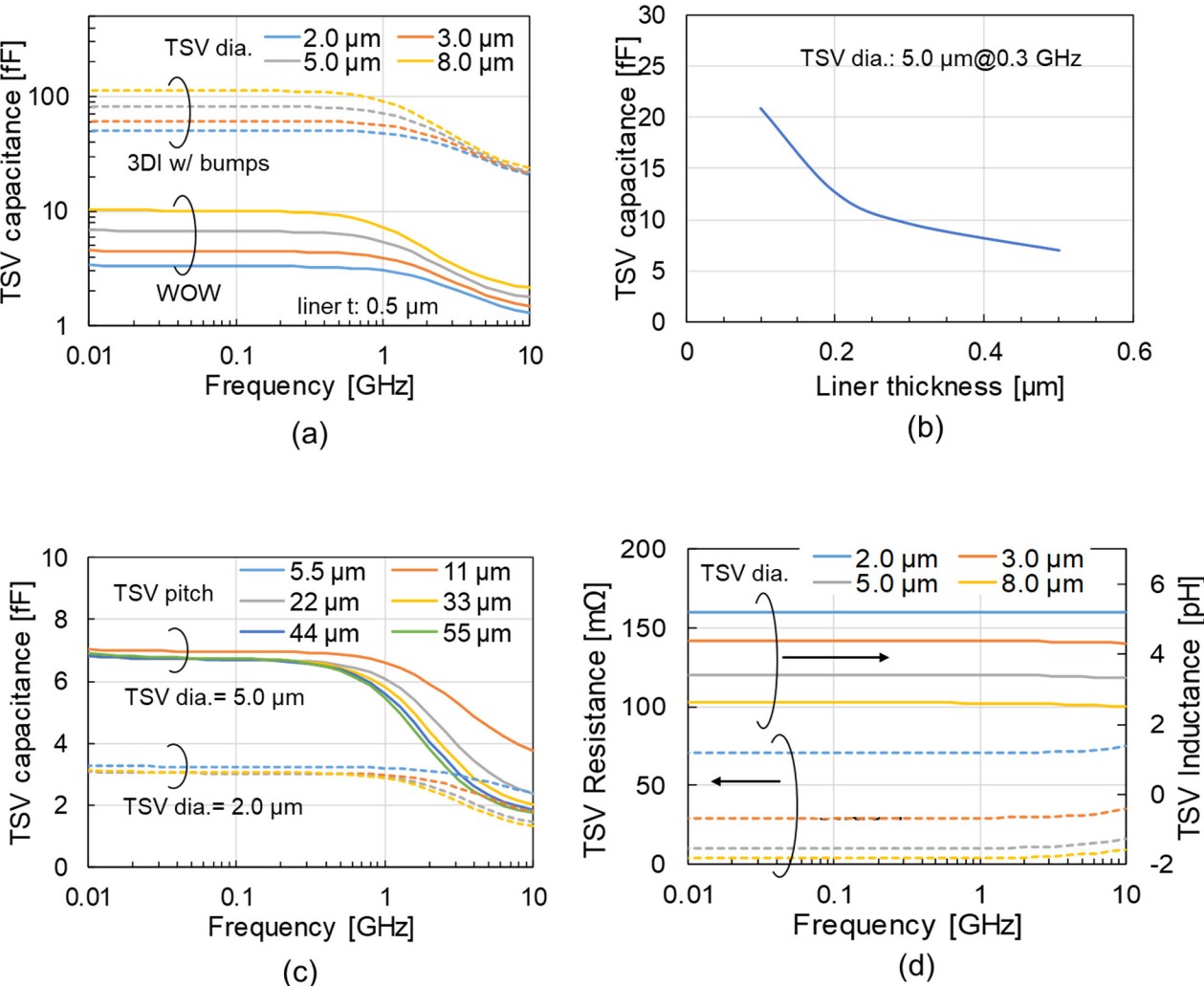

**Figure 31.** Calculation results of TSV capacitance: (**a**) frequency characteristics of capacitance, (**b**) impact of liner thickness, (**c**) impact of TSV pitch and (**d**) frequency characteristics of resistance and inductance.

Circuit simulation was used to estimate the power consumption of the I/O circuit. An eye diagram was calculated, and the I/O current that satisfies the eye mask was determined [34]. The structure of BBCube is presented in Figure 32 and a block diagram of the simulation is shown in Figure 33a. By utilizing the capabilities of the dense TSVs, the data rate was set at only 800 Mb/s, which is lower than the HBM2E of 3.2 Gb/s. Nine DRAM die was stacked in case of BBCube. An additional die was used to implement novel 3D-based redundancy. The I/O current was assumed to be proportional to the output resistance and capacitance of the I/O buffer circuit. The circuit parasitic capacitances $C_o$ (driver output capacitance), $C_d$ (capacitance at dropping point of TSV) and $C_{in}$ (receiver input capacitance), assumed to be proportional to the driver output resistance $R_o$. $T_1$ was an s-parameter model of the TSV, calculated by 3D EM analysis. The HBM and BBCube results

are compared in Figure 33b. BBCube achieved 30-times higher I/O power efficiency. In the stacked memory as a whole, BBCube could realize a four-times higher bandwidth with only 13% of the I/O power consumption compered to HBM, as shown in Figure 34. The TSV area in the DRAM die became 64% using 12 μm pitch, and the I/O circuit area becomes about 1/30th. Furthermore, as the process technology matures, the next generation (Gen2) should be able to achieve 32-times higher bandwidth with the same I/O power consumption as HBM.

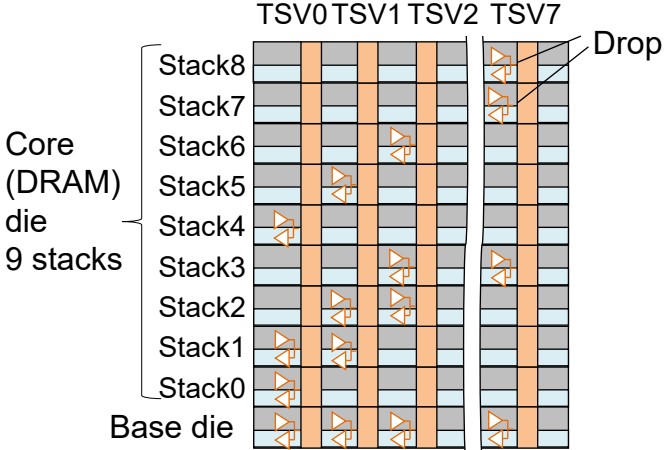

**Figure 32.** Configuration of BBCube.

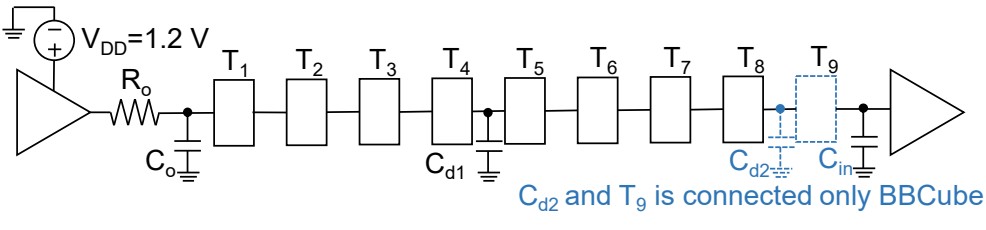

(**a**)

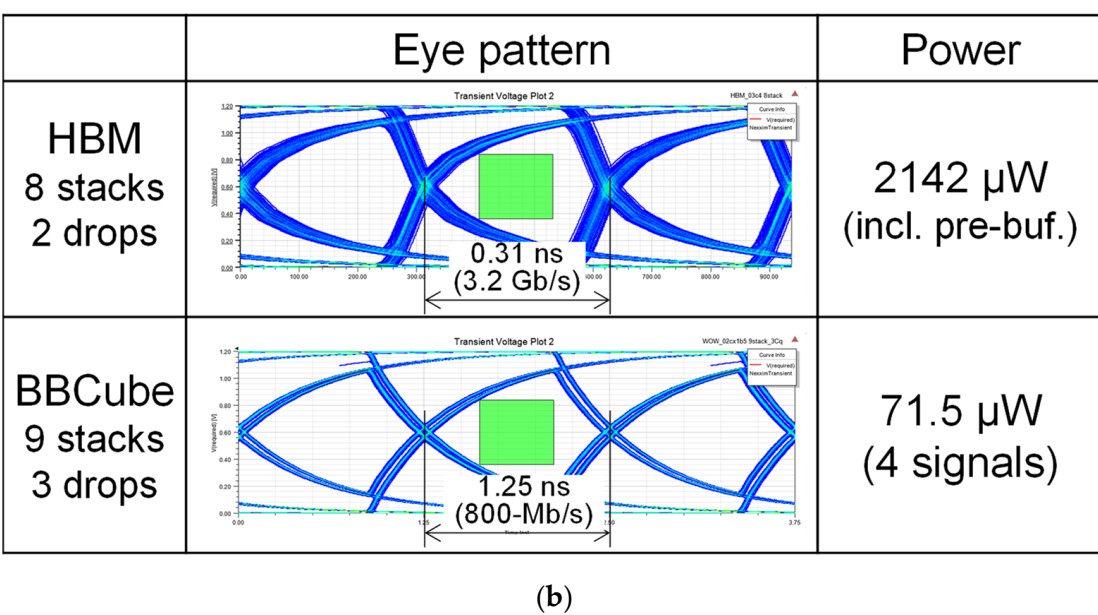

(**b**)

**Figure 33.** Eye diagram and data transfer power within a TSV: (**a**) block diagram of simulation and (**b**) comparison between HBM and BBCube.

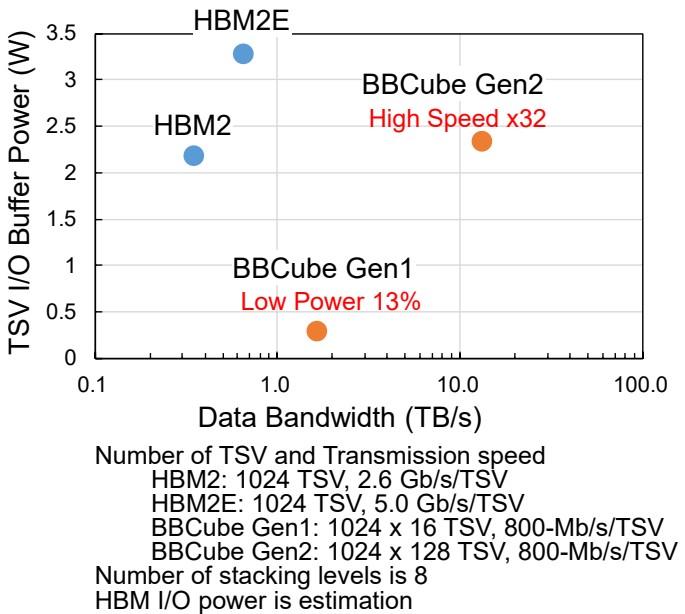

**Figure 34.** A comparison of the data bandwidth and TSV I/O power consumption between HBM and BBCube, where HBM2 [85], HBM2E [86], repectively.

*8.2. Thermal Characteristics of BBCube*

The temperature of a DRAM cell influences its retention time and limits the number of stacks [90]. Therefore, a thermal analysis of stacked DRAMs was performed. The TSVs in the BBCube were connected directly to the bottom die, whereas with conventional 3DI, it is necessary to put a solder and a BEOL layer between the TSVs, which increases the thermal resistance. The thermal resistance in the BBCube case was 1/4th that of conventional 3DI [91]. Figure 35 shows the temperature difference between the top of the stacked DRAM, which was at room temperature, and the highest temperature part of the DRAM cell. Under the stacked DRAM, a base die with the same power consumption was placed in both the HBM and BBCube. For the BBCube with 9 stacks, the difference of the DRAM cell temperature was 8.3 °C due to the low thermal resistance. Even when 34 dies were stacked, the difference in temperature in BBCube was 16 °C, which was about two-thirds that of HBM with 8 stacks. BBCube allowed stacking of 4-times more dies than HBM. This allowed the memory capacity to reach 64 GB using 16 Gb DRAM dies.

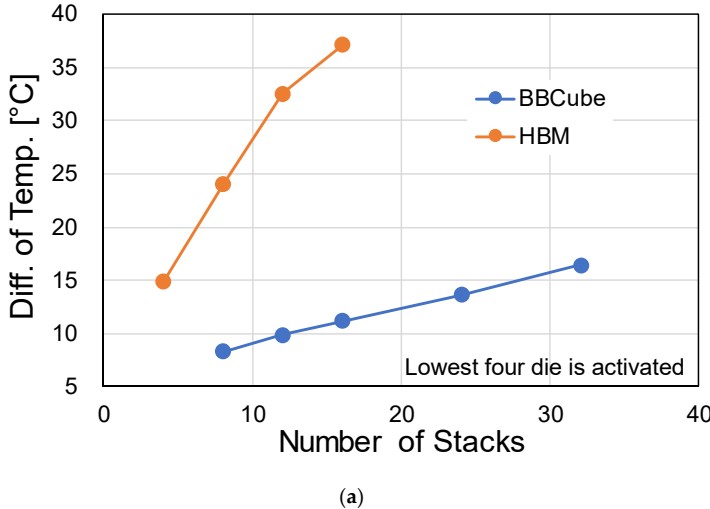

(a)

**Figure 35.** *Cont.*

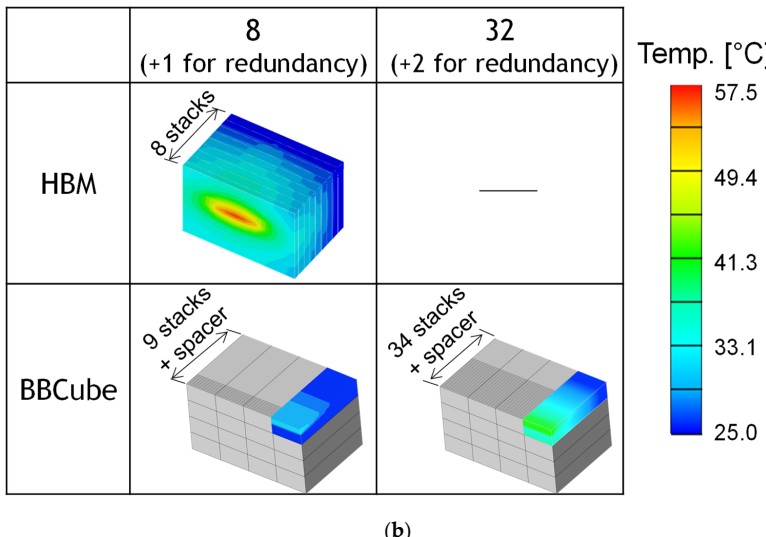

(**b**)

**Figure 35.** Thermal simulation results: (**a**) the temperature difference between the top of the stacked dies, which is set to room temperature, and the highest temperature part of the DRAM cell and (**b**) heat distribution.

### 8.3. Competitive BBCube DRAM Structure

The competitive BBCube DRAM structure is one that enables 8-die stacking with bumpless TSVs. By increasing the number of channels and lowering the TSV impedance, ultra-high bandwidths of 1, 4, and 8 TB/s should be achievable, as illustrated in Figure 36.

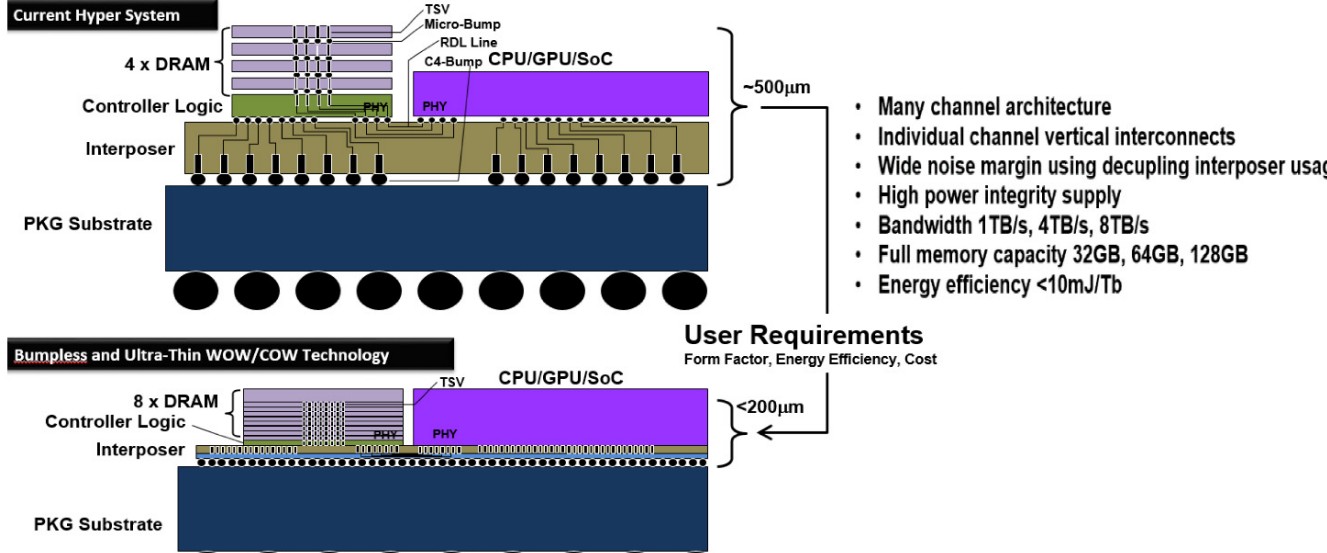

**Figure 36.** A comparison of conventional DRAM stack such as HBM and BBCube memory for 2.5D system. Competitive 3D structure accompanied with lowering system height and large memory capacity is anticipated.

Figure 37 shows the HBM data bandwidth roadmap. By realizing the parallelism enhancement due to the increase in the number of I/O's, the bandwidth of the HBM, which had no bumps, was expected to be ever-increasing. As for the I/O power consumption, the first target of the bumpless HBM was one-thirtieth that of the current HBM2 [28], as shown in Figure 38.

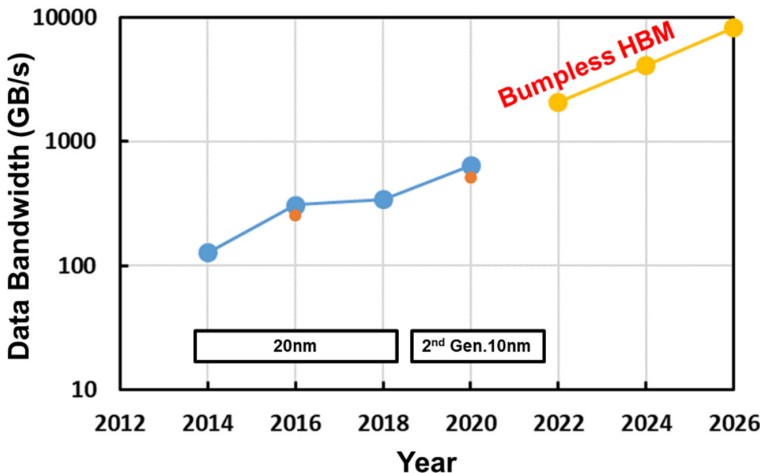

**Figure 37.** Data bandwidth roadmap [82–87].

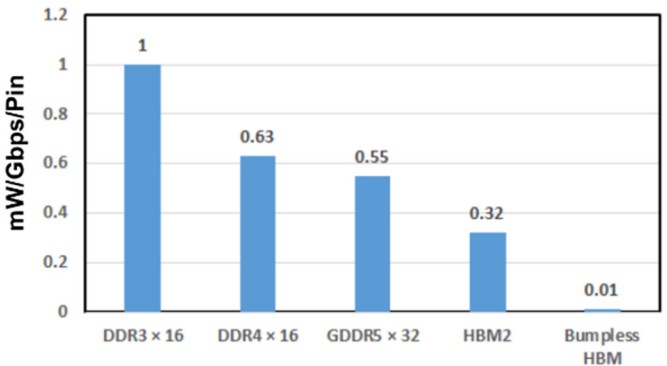

**Figure 38.** I/O power efficiency.

Figure 39 compares HBM2 with bumps [31–33] and bumpless HBM with respect to their data bandwidth and I/O buffer power, according as the number of I/O's [28,29,34,92,93].

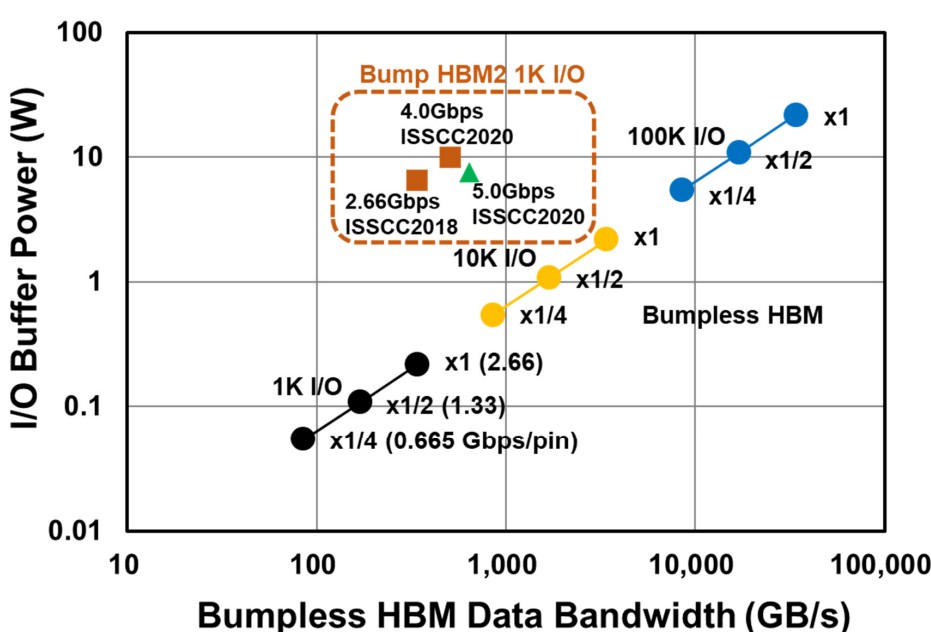

**Figure 39.** Data bandwidth and I/O buffer power comparison [85–87].

Bumpless HBM can achieve an ultra-high data bandwidth by increasing the I/O number to 1 K, 10 K and 100 K, and can lower the I/O buffer power to 1/2 or 1/4 by reducing the I/O pin frequency with a four-phase shielded I/O scheme, as illustrated in Figure 40. The great advantage of this scheme is validated in Figure 33.

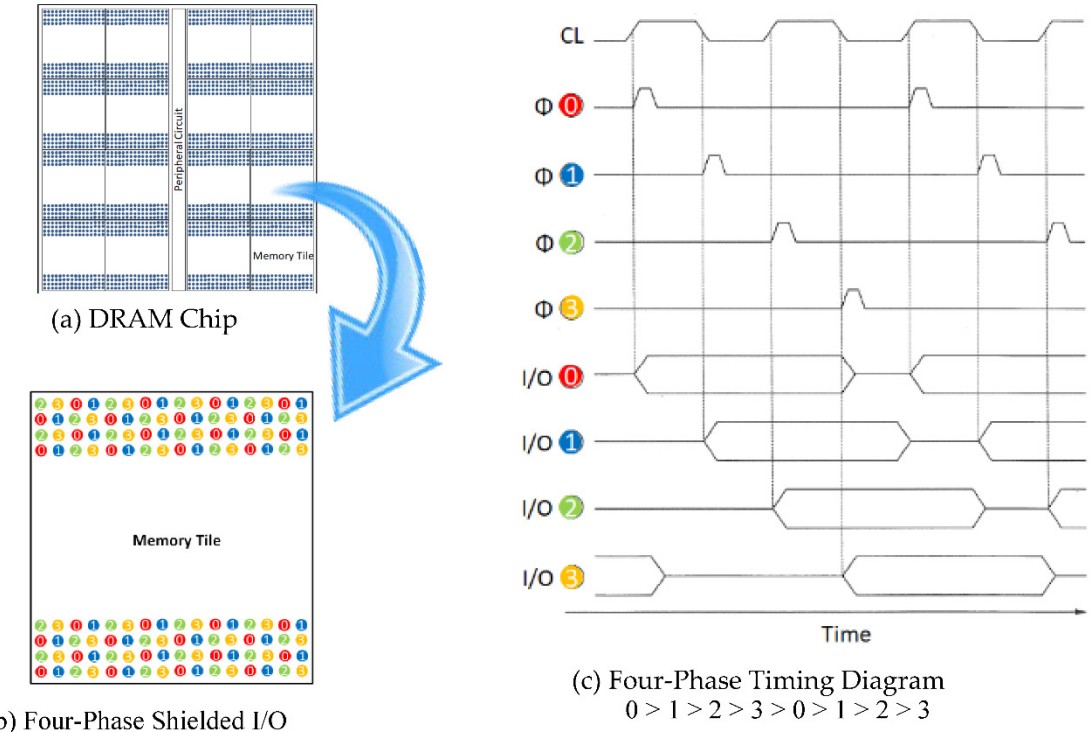

(a) DRAM Chip

(b) Four-Phase Shielded I/O

(c) Four-Phase Timing Diagram
0 > 1 > 2 > 3 > 0 > 1 > 2 > 3

**Figure 40.** Four-Phase Shielded I/O Scheme.

### 9. BBCube NAND

*9.1. Limitations of Stacked WL Tiers in 3D NAND Chip*

i.　If 64 tiers by one etching shot is limited by the highest aspect ratio, 512 tiers would require 8 times cell process, such as $64 \times 8$. Therefore, there is a large heat budget to enhance the source/drain diffusion of the transistors for both a CMOS Under Array (CUA) and a CMOS Next Array (CNA). As a result, the peripheral transistors would be very large, and their performance would be degraded.

ii.　If the number of cells per string should increase to 128, 256, and beyond, the cell current would be very small, so that random page access would become slower.

iii.　In the case of (2), both the page count and block size must be large, which would be user unfriendly for reprogramming, such as data copying and moving.

iv.　A solution to the issues in (2) and (3) would be a multiple vertical bitline architecture, but this would make routing and wiring difficult within the tight XY bitline pitch.

v.　The number of high-voltage transistors for NAND string drivers must also increase according as the number of stacked WL tiers, which would occupy a huge Si area in spite of the CUA structure.

Thereby, the number of the stacked WL tiers would be limited at the 256 tiers, which is produced by $128 \times 2$ of the twice cell process.

*9.2. Vertical Bitline Architecture*

Figure 41 shows the stacked 3D NAND chip in the case of one 3D NAND chip. Figure 42 illustrates four stacked 3D NAND chips which are connected with a vertical bitline (BL), as well as bumpless TSVs [92,93].

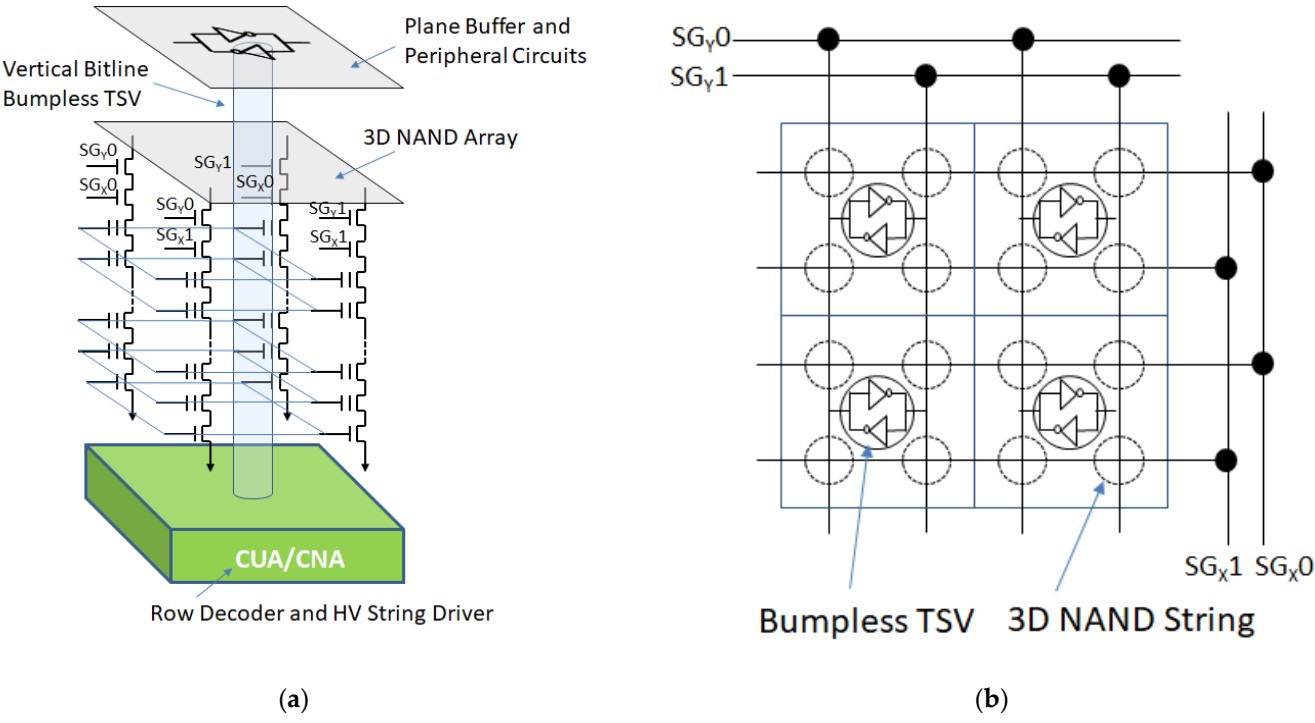

**Figure 41.** Vertical bitline architecture. (**a**) bird's eye view; (**b**) top view.

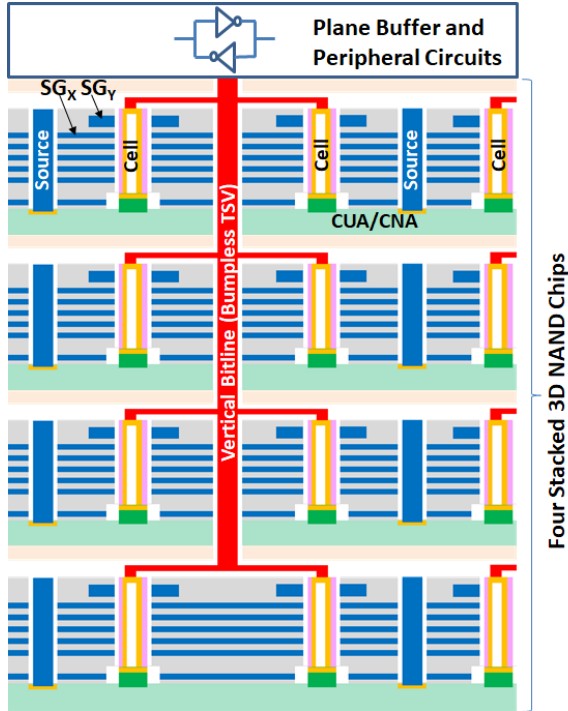

**Figure 42.** Four 3D NAND chips connected through Vertical BL.

### 9.3. Word Plate Access NAND

Toshiba proposed an original 3D NAND device for the first time [94]. Multiple CGs and Lower SGs were merged in each plate to reduce the number of HV-driver transistors, as well as to tighten the pillar pitch. In this study, the polysilicon word plate was replaced by a damascene tungsten metal gate, but the basic 3D NAND structure of the merged multiple CGs and Lower SG was the same as the original one. Figure 43 presents a future 3D stacked

memory design [95]. By increasing the number of TSVs, a part of a peripheral circuit of the first memory chip can be located in the second memory chip. A stacked DRAM combined with a NAND flash memory is illustrated below as an example.

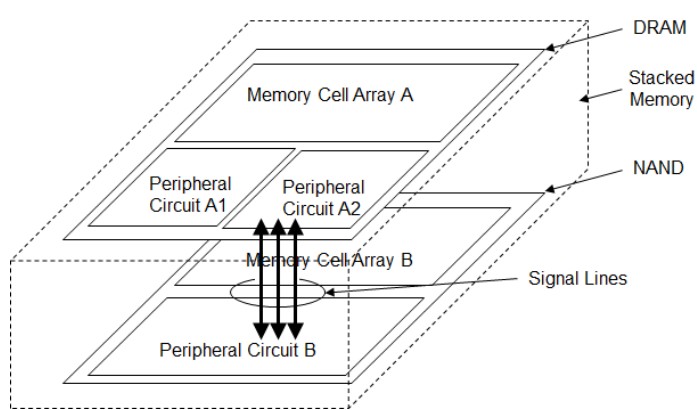

**Figure 43.** Future 3D stacked memory system [95].

Figure 44 shows a proposed BBCube NAND with multiple BL layers. Thanks to the original architecture of 3D NAND, a word line is expanded to a word plate, so that the 3D NAND can be read and programmed *by plane* instead of *by line*.

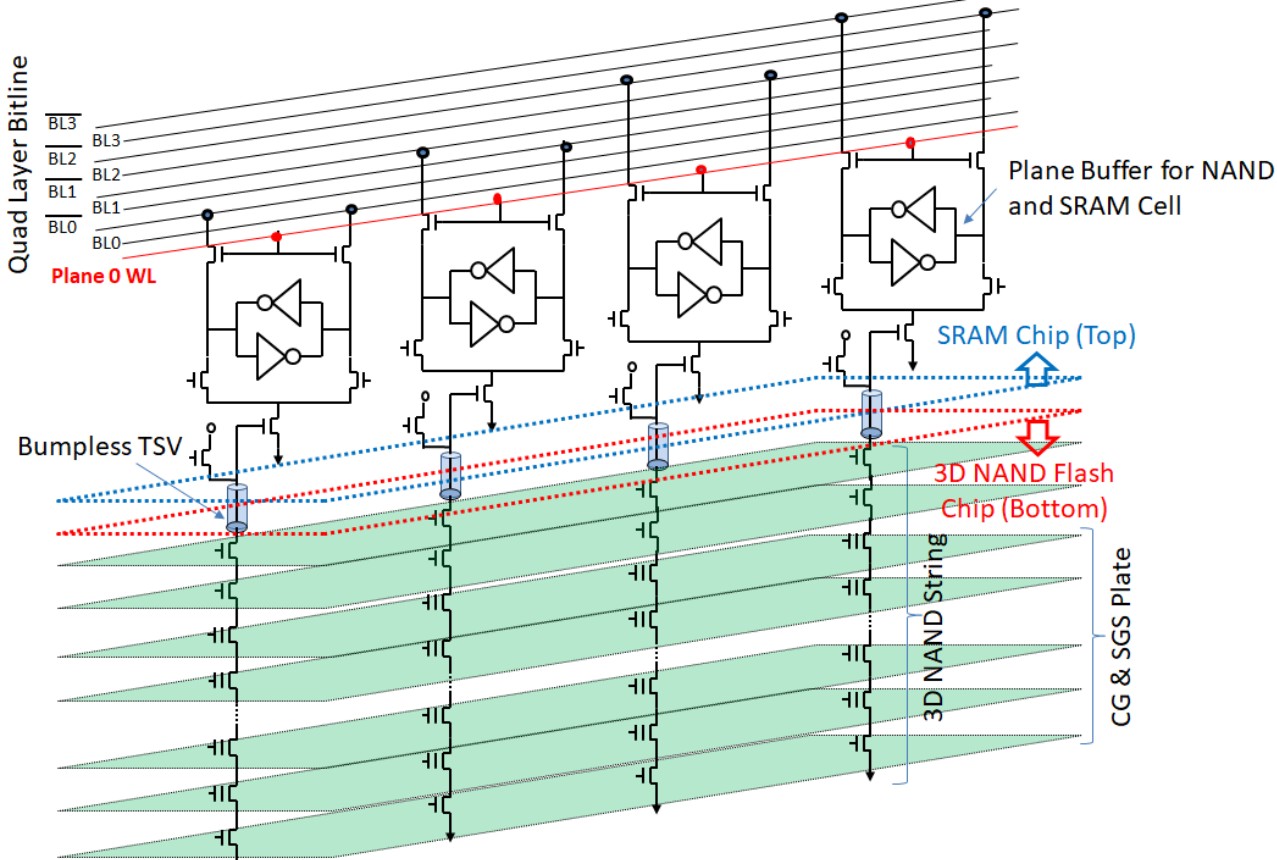

**Figure 44.** Circuit diagram of BBCube NAND composed of two chips.

## 10. BBCube Memory Application

WOW technology with bumpless interconnects using TSVs for three-dimensional stacking in wafer form has been described. An optimized thinned wafer thickness of 4 µm

can increase the number of TSVs per chip with the fine pitch of the TSVs and can reduce the impedance of the TSV interconnects with no bumps. Therefore, an even higher-speed and higher-density HBM, namely the BBCube DRAM, can be realized with the four-phase shielded I/O scheme. Additionally, the BBCube NAND with the vertical BL architecture, which can be read and programmed *by plane* instead of *by line* by using the bumpless TSV, has been proposed. The BBCube DRAM for RAM and the BBCube NAND for ROM are sister memories with the high bandwidth.

As the number of the stacked memory chips is increased, the total memory density should be huge, similar to an enterprise. Therefore, an AI robotic bee, as an example, that can be used as a human assistant, which has a CPU, ultra-small enterprise, BBCube DRAM, BBCube NAND, and sensors, should be eventually realized in 50 mm$^3$ with 0.5 mW power consumption, as proposed in Figure 45.

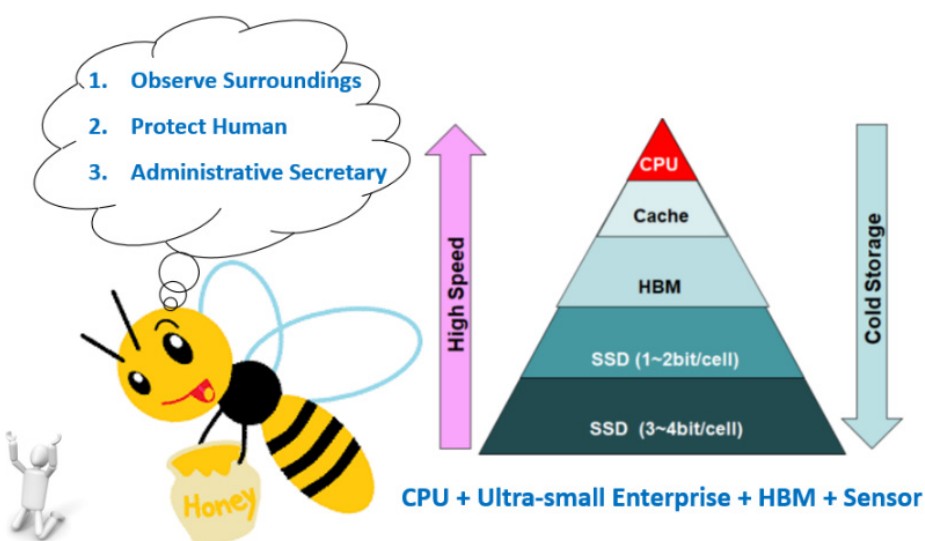

**Figure 45.** AI Robotic Bee (50 mm$^3$, 0.5 mW) as a human assistant.

## 11. Introduction to 3D Redundancy

In this section, our research motivation is to realize wafer-level fabrication, by which we can provide higher density and lower impedance TSVs with excellent heat conductivity, as discussed in this paper.

Figure 46 shows the configuration of the stacked DRAM system in BBCube generation one. It consists of 8 stacked dies, with one extra die for 3D redundancy, which will be discussed later. Each die is equipped with 16 tiles, and each tile has 4 or more banks. These tiles are memory arrays with 1024 I/Os vertically connected by TSVs. Therefore, massively high parallelism of 16 k I/Os was realized. Within each bank, sub-arrays of DRAM cells with extra sub-arrays are provided to perform intra die redundancy of a layer-by-layer scheme. We called this two-dimensional (2D) redundancy.

The superior properties of TSVs, such as lower impedance and higher heat conductance compared with existing micro-bump structures, originate from a technique for ultra-thinning Si substrates [34]. The quality and reliability of TSVs are supported by a copper dual damascene (DD) technique, which is very common for front end of line (FEOL) processes in device manufacturing. Since devices are already placed on wafers with the placement accuracy of lithography tools, wafer form fabrication is potentially capable of achieving layer-to-layer alignment with nm-level precision. These techniques have been completely proven in the manufacturing processes of devices and materials, such as CMOS image sensors (CIS), silicon on insulator (SOI) substrates, and microprocessors

(MPUs) [96–98]. Wafer form fabrication is the key to utilizing these techniques, and to enjoying the benefits of the maturity of manufacturing equipment.

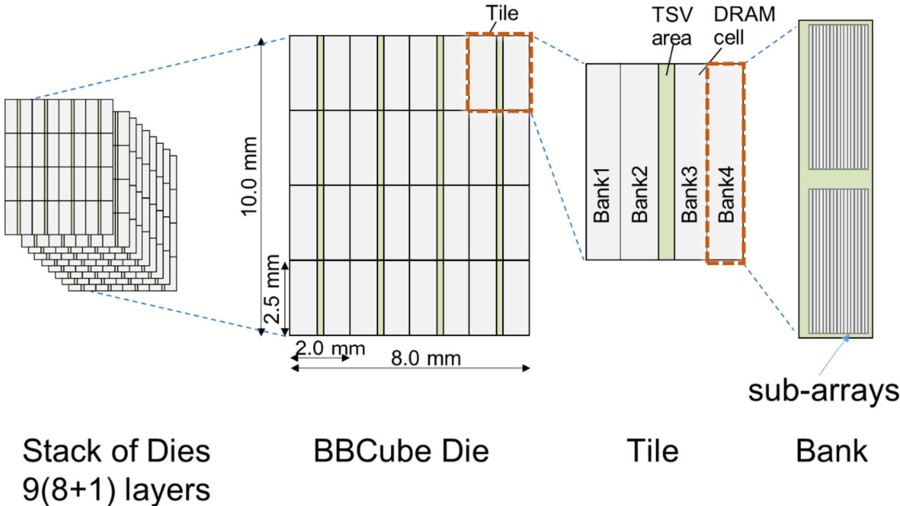

**Figure 46.** Configuration and structural hierarchy of the stacked DRAM system, BBCube. Banks consisted of sub-arrays for 2D redundancy, and stacked dies of 9 layers for 3D redundancy are illustrated.

To realize wafer form fabrication, it is essential to investigate defect management design, especially for random defects, since the probability of randomly defective portions being included in the module stack cannot be eliminated, as illustrated in Figure 47a. On the other hand, conventional KGD processes performed wafer testing, so that it is possible to stack defect free dies, as shown in Figure 47b. By simple arithmetic, the stacked device yield of the KGD process, $Y_{3D}^{KGD}$, was equal to the wafer test yield, $Y_{device}$, as defined in Equation (4). Besides, the yield of the wafer stacking case, $Y_{3D}^{WoW}$, was calculated from the wafer test yield to the power of the number of stacked layers, k, without any remedy, as expressed in Equation (5).

$$Y_{3D}^{KGD} \equiv Y_{device} \tag{4}$$

$$Y_{3D}^{WoW} = (Y_{device})^k \tag{5}$$

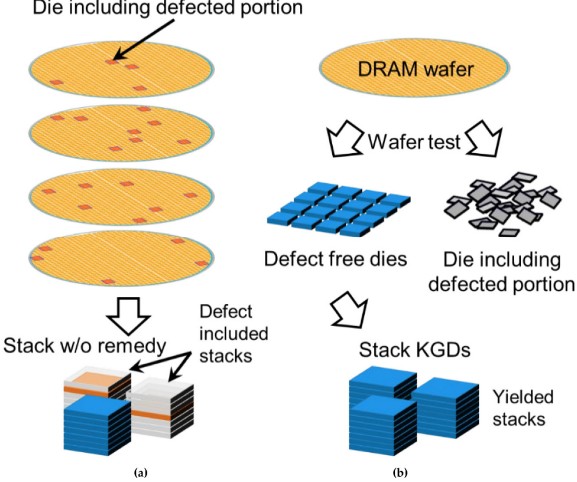

**Figure 47.** (**a**) For the wafer form fabrication, the probability of randomly defective portions being included in the module stack cannot be eliminated. (**b**) On the other hand, conventional KGD process includes wafer testing, so that it is possible to stack defect-free dies.

Here, we describe 3D redundancy [99,100] as applied to the configuration of the first generation BBCube. Using a general stacked DRAM system configuration, we illustrated the techniques constituting 3D redundancy in detail. It is apparent that these techniques are more practical and rational.

### 11.1. Method of 3D Redundancy

11.1.1. Typical Configuration of Stacked Synchronous DRAM Systems

Figure 48 shows a schematic diagram of a typical configuration of stacked DRAM devices [101], in which the circuit design hierarchy is the same as that of BBCube. In general, each die consists of banks [85]. These banks include sub-arrays, with redundant sub-array(s) to replace defective sub-array(s) while performing 2D redundancy. This 2D redundancy was carried out on a die-by-die basis within each layer of the stack. Recent DRAM devices were provided with extra cell arrays occupying 10% to 20% of the total area to reduce bit error rate (BER), in both cases of block sparing type redundancy and error check and correction (ECC) [86]. In this paper, we used a simple sub-array sparing model to be discussed later, to evaluate the area overhead.

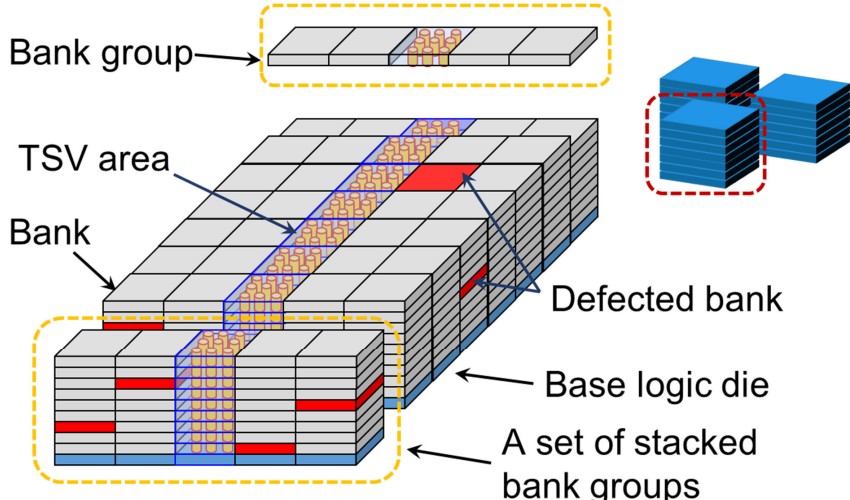

**Figure 48.** Schematic diagram of typical configuration of stacked DRAM devices. The circuit design hierarchy of BBCube is common. Each die consists of banks. These banks include not only sub-arrays for memory capacity, but also redundant sub-arrays to replace defective sub-array(s) to perform 2D redundancy.

When the 2D redundancy fail, the defective banks remain in the dies. In the case of the KGD process on the other hand, the dies were disposed of, which was in vain [101]. The TSV area connected neighboring banks to other layers, to avoid longer intra-die wiring [85–87]. The neighboring banks associated with the same TSV "digits" were grouped into bank groups, which corresponded to tiles in the case of BBCube. With WOW technology, the calculated delay basis distance in the z-direction between neighboring layers was approximately 30 μm. Therefore, in a set of stacked bank groups, banks were mutually compatible and replaceable with each other. It is possible to prefetch data from banks in different layers through the bypassing of vertical global wiring (Copper TSVs) in front of the buffer circuit block.

A base logic die includes a test circuit, a high-speed interface (HSIF), and a DRAM control circuit.

11.1.2. Techniques for 3D Redundancy

Based upon the typical configuration described above, we introduced a 3D redundancy, which consisted of three techniques, and a derivative extension at the sub-array level. The target was to provide the maximum number of stacked devices from the total fabricated

DRAM silicon area. Note that 3D redundancy is combined with 2D redundancy to reduce defect density, so as to be applicable to a vertically replaceable memory block architecture.

### 11.1.3. Layer Addition to Cover Circuit Resources

As illustrated so far, if defective banks result from the defect rate exceeding the 2D redundancy capability, the total number of non-defective banks is not enough for stack device operation. To enable repair of such devices, we added an extra layer to supplement the required number of non-defective banks, which is illustrated in Figure 49a. Note that this is not adding redundant cell arrays. The supplementally stacked die(s) were completely compatible with other dies inside the stacked layers below.

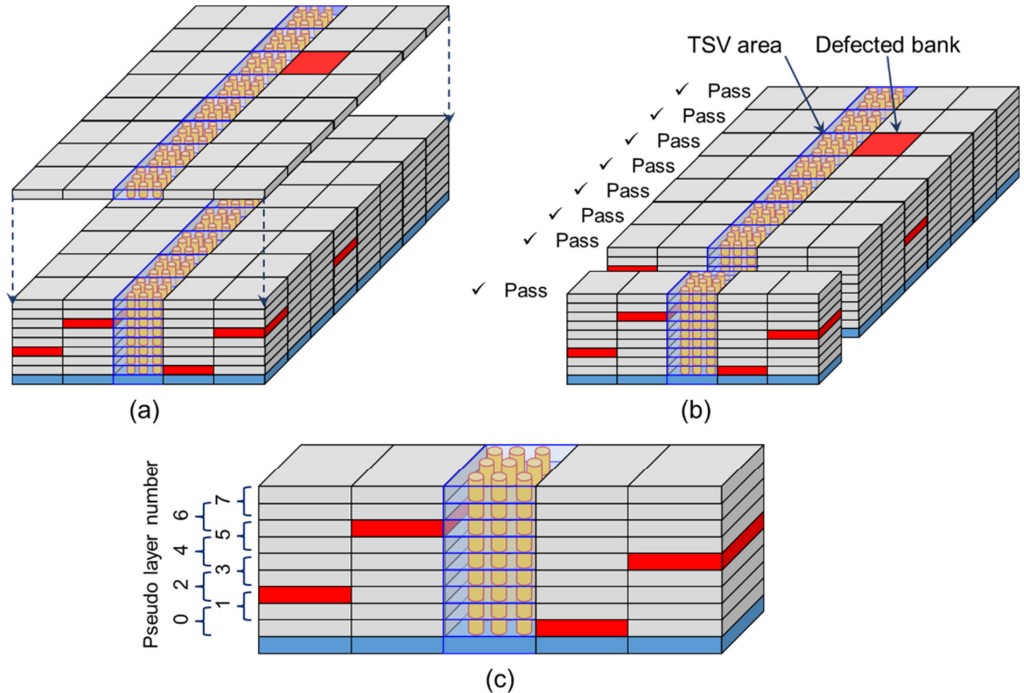

**Figure 49.** 3D redundancy techniques for: (**a**) layer addition to cover circuit resources, (**b**) bank replacement among a set of stacked bank groups, and (**c**) pseudo layer-by-layer operation.

### 11.1.4. Bank Replacement within a Set of Stacked Bank Groups

The next technique was to replace banks within a set of stacked bank groups. The banks, which belong to TSV "digits" were grouped together as a bank group. These bank groups were stacked as a set of stacked bank groups. Basically, banks supported a closely located set of TSV "digits", to avoid an increase in internal wiring capacitance, as illustrated in the previous section. Therefore, we were able to replace defective banks within individual sets of stacked bank groups to provide the overall functionality of individual TSV "digits". Note that, in different TSV "digits" (set of bank groups), the combination of selected functionable banks can be different from those of others. This scheme gives us another degree of freedom for memory repair optimization. Eventually, the entire functionality of all TSV "digits" will be individually accomplished, as indicated in Figure 49b.

### 11.1.5. Quasi Layer-by-Layer Operation

So far, in this discussion, it has been assumed that all stacked-die layers are connected to TSV equally. However, connected I/O transistors behave as load capacitance, even if they are not in use. TSVs should be connected to a minimum number of stacked-die layers. The third technique for achieving 3D redundancy is quasi layer-by-layer operation. For a functional stacked device, in a set of bank groups, the maximum possible number of defective banks must be equal to the number of banks included in the added extra layer or layers. This means that if we allocate twice as many layers as extra added layer(s) to

a pseudo layer, we can obtain quasi layer-by-layer operation, as illustrated in Figure 49c. With this technique, logical layer can be defined as same as conventional stacked DRAM devices. In the case of Figure 49c, it is possible to share total required I/Os of data bus in 9 layers instead of 8 layers. Therefore, it is possible to reduce required silicon area of I/O transistors for a layer [102].

### 11.1.6. Derivative Extension Case: 3D Redundancy at Sub-Array Level

In the discussion so far, we have assumed the typical configuration of stacked DRAM systems such as HBM. In the BBCube case, each bank provided 1024 bit-wide I/Os in a tile, so that mutual compatibility within the set of the bank groups (tiles) was guaranteed. However, the tiles of BBCube, which were already highly fine-grained and partitioned into narrower pseudo banks from a 1024 bit-wide bank should be considered for better energy efficiency [103]. In such cases, each bank group (tile) may involve only one bank, or one pseudo bank, as illustrated in Figure 50a. Such a case makes 3D redundancy much less effective. We investigated whether we could use the sub-array level, which is the next level in the hierarchy below the bank level in typical DRAM and BBCube configurations.

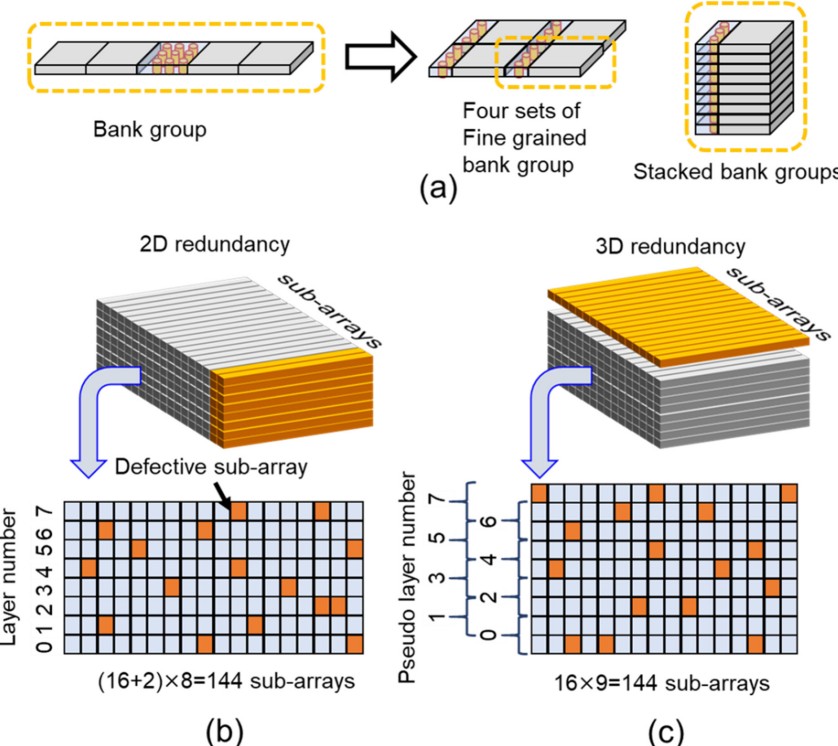

**Figure 50.** (**a**) Cases where each bank group (tile) may involve only one bank, or one pseudo bank. (**b**) In the case where 2 more redundant sub-arrays give near 100% yield, more than 127 non-defective sub-arrays are included in 144 sub-arrays. (**c**) In case of the same defective rate as (**b**), stack of single bank tiles with 1 extra layer tile includes 16 defective sub-arrays at most.

The sub-array level is used for 2D redundancy, to achieve excellent yield for the KGD process. With a certain amount of area penalty, near 100% yield can be obtained, as illustrated in Figure 50b.

In the case where two more redundant sub-arrays give near 100% yield, the maximum number of defective sub-arrays in the stack of tiles must be 16. Thus, (16 + 2) ×8 = 144 sub-arrays must include 128 fine sub-arrays, as shown in Figure 50b. With a different configuration, 16 × 9 = 144 sub-arrays also must include 128 fine sub-arrays. This means that two vertically neighboring physical layers should include 16 or fewer defective sub-arrays. As a result, assignment of 8 pseudo layers out of 9 physical layers is possible, as illustrated in Figure 50c.

This "two more sub-arrays are enough" situation is realized by yield improvement activities and more nested 2D redundancy. The data transferred from replaced sub-arrays needs to be bypassed across physical layers, before the data multiplexers that provide bank data to the I/O buffers.

Accordingly, sub-array level 3D redundancy is feasible, and the bank configuration in a tile should be flexible to achieve energy efficiency optimization.

### 11.1.7. Parameter Definition

The definitions of parameters for the yield calculation are illustrated in Figure 51. In the case of BBCube, the number of tiles corresponded to the number of bank groups of typical DRAM systems in the figure. The scheme in which stacked bank groups support TSV "digits" becomes clearer when considered on a tile-by-tile basis.

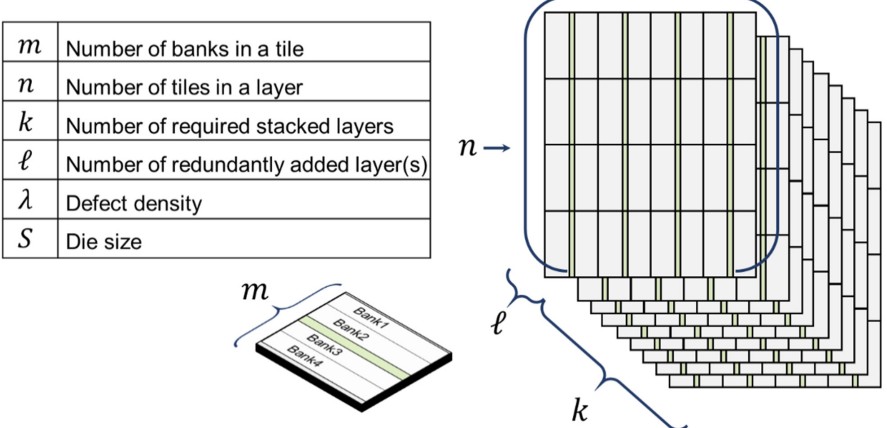

| $m$ | Number of banks in a tile |
|---|---|
| $n$ | Number of tiles in a layer |
| $k$ | Number of required stacked layers |
| $\ell$ | Number of redundantly added layer(s) |
| $\lambda$ | Defect density |
| $S$ | Die size |

**Figure 51.** BBCube stack configuration and symbols for calculation.

When performing 2D redundancy, it is assumed that each bank includes 16 sub-arrays with redundant sub-arrays. Therefore, one extra sub-array incurs a 6.25% (i.e., 1/16) area penalty. The calculation was carried out with a simple sub-array replacement model.

### 11.1.8. Yield Calculation

In this paper, a Poisson distribution model was assumed as for the random defect yield model, which is consistent with the discussion below on intrinsic random defects, and does not result in a loss of generality of the whole discussion. Device yield, $Y_{device}$, is expressed by:

$$Y_{device} = Y_s \times Y_r \rightarrow Y_{device} \equiv Y_r = \exp(-\lambda S) \tag{6}$$

$$Y_{Bank} = \exp\left(-\frac{\lambda S}{n \times m}\right) = Y_r^{\left(\frac{1}{n \times m}\right)}, \; Y \equiv Y_{Bank} \tag{7}$$

where $Y_s$ is systematic yield, $Y_r$ is random defect yield, S is the die size, and $\lambda$ represents the area density of random defects [104].

The systematic yield $Y_s$ is linked to various root causes from its definition as "systematic". They are identified and dealt with in the scope of process development, yield improvement activities, design for manufacturing (DFM) techniques, and big data analytics from manufacturing processes [105]. These efforts also target randomly distributed defects, which extrinsically cause random defect yield loss.

On the other hand, there are random defects that cannot be sufficiently reduced, even with intensely run yield improvement activities. For example, the variation of DRAM cell retention time due to impurity profile fluctuations can be minimized, by engineering efforts, in controllable portions of the processes, but there are remaining portions where intrinsic fluctuations exist. Fabrication engineers are struggling to achieve yield improvements, but sometimes encounter non-visible defects. This type of randomness is an essential barrier

to yield improvement for both WOW devices and leading-edge devices, because we have already entered an era in which the number of atoms should be considered as an index of pattern pitch [106].

Problems that have root causes can be solved by eliminating them. Therefore, we assume that $Y_s$ is close enough to "1" and concentrate on intrinsic $Y_r$.

In this analysis, we used the term "KGD case", which means a process that involves testing, and screened die stacking. It may be possible to apply 3D redundancy for stacked diced devices with a micro-bump structure. In that case, the calculation result will be the same between "WOW" and "stacked diced device" cases. An aim to identify something that could act as a benchmark led us to a comparison of the fabrication process differences. In this paper, we focus on the differences in redundancy procedures, and we do not go into the differences in the processes. We assumed the same 3D integration process yield of 100% for both WOW and KGD cases. For performance evaluation in other sections, we were able to deal with the differences in device structures.

The following Equation (8) presents the model for the yield of BBCube, $Y_{3D}$, with the 3D redundancy scheme illustrated in this paper. The parameters are described in Figure 51. When the single layer die yield (wafer test yield), $Y_{device}$, is given by Equation (6) and the bank yield is calculated from Equation (7), the BBCube yield, $Y_{3D}$, can be expressed as:

$$Y_{3D} = \left[ \sum_{i=k \times m}^{(k+\ell) \times m} \left\{ \left( {}_{(k+\ell) \times m}C_i \right) \cdot Y^i \cdot (1 - Y)^{(k+\ell) \times m - i} \right\} \right]^n \tag{8}$$

$$Y_{3D}^{KGD} \equiv Y_{device} \tag{9}$$

In the formula, the total yield of a tile is calculated by summing all products of bank yield and defect rate weighted by number of its combination. The random defect yield of the targeted tile is calculated as the term in brackets in Equation (8), so that the yield of the whole BBCube system can be obtained as the tile yield to the power of the number of tiles. To compare with BBCube yield, $Y_{3D}^{KGD}$ by the KGD process is equal to $Y_{device}$ itself, because it is possible to select a functional die by testing, which is expressed as in the definition in Equation (9), which is the same as Equation (4).

*11.2. Results and Discussion*

11.2.1. Results of BBCube Yield

Our goal was to yield the maximum number of stacked devices (BBCube) from the total fabricated silicon wafer area of the device.

As indicated in Figure 52a, BBCube fabricated by the WOW process showed better yield than that of the KGD stacking case, for all single layer die yields $Y_{device}$. The reason is as follows. For the KGD case, if the required number of banks is not available in a silicon die, the die area is wasted. On the contrary, for the WOW case, when there is an error in banks, the necessary number of banks can be substituted from other layers in the stack of bank groups. Therefore, when the single layer die yield $Y_{device}$ is greater than or equal to 50%, the BBCube yield becomes more than 99% with 3D redundancy. If we want to achieve such excellent yield with only 2D redundancy, we must prepare more redundant sub-arrays. The area penalty evaluation of 2D redundancy is shown in Figure 52b.

Figure 52b illustrates, when a die yield without redundancy is given, how much area penalty is required to achieve the target yield by 2D redundancy. The die yield without redundancy is sometimes called the "perfect yield". The target yield cases evaluated are greater than 50% (">50%"), ">99.5%" and ">99.99%".

For productivity comparison, we needed to consider the area penalty of redundancy schemes. In the case of 3D redundancy, 9 wafers were consumed to realize the device function of 8 layers. Therefore, the area penalty of 3D redundancy was 12.5%, without a consideration of the area penalty of 2D redundancy, so that, in the case where the single layer die yield $Y_{device}$ is greater than 87.5%, the KGD process seemed to be more productive.

However, to realize such an excellent single layer die yield $Y_{device}$, an area penalty of 12.5% or more was necessary for 2D redundancy only. Moreover, for almost the entire practical range of die yield without redundancy, to achieve a target yield of ">99.5%," 2D redundancy needed 12.5% or more die area than the case of a target yield of ">50%," as illustrated in Figure 52b. For 3D redundancy, a target yield of ">50%" was enough to achieve a single layer die yield $Y_{device}$ of more than 99% of the BBCube yield. Therefore, 3D redundancy realizes better productivity, even under such conditions.

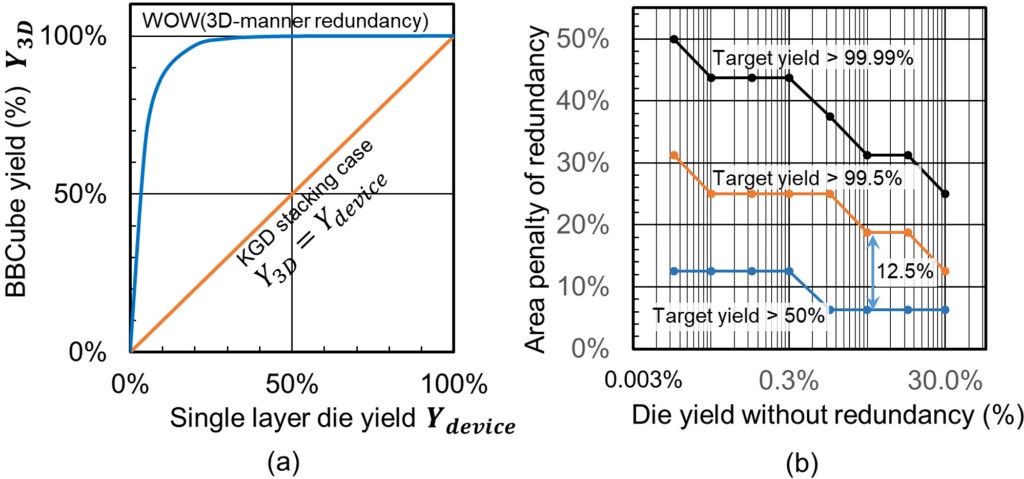

(a)  (b)

**Figure 52.** (**a**) BBCube yield comparison between WOW and KGD cases. (**b**) Area penalty of 2D redundancy required to achieve target yields of ">50%", ">99.5%" and ">99.99%".

Figure 53 shows a yield comparison of the cases for BBCube when more layers are aggressively stacked. Cases in which 9 (8 + 1), 17 (16 + 1), and 33 (32 + 1) layer are stacked are shown. Even in the case of 33 layers, with a single layer die yield $Y_{device}$ of more than 80%, more than 99% BBCube yield was achieved, which indicates that 3D redundancy can support wafer form fabrication up to such an aggressive number of stacked layers. The portion for the added layer overhead for 3D redundancy is lowered to 3.125% (1/32) in this case.

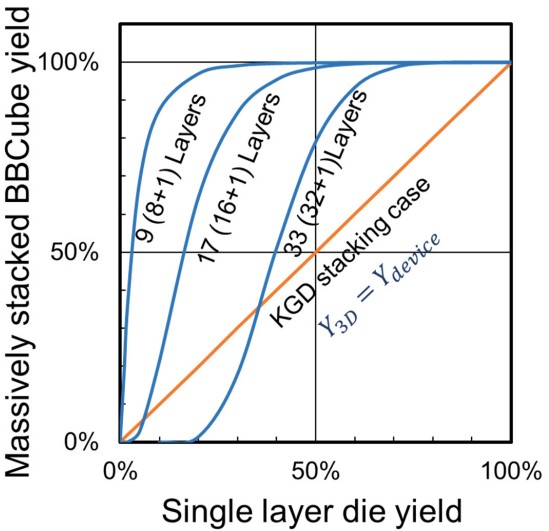

**Figure 53.** Yield comparison of the cases for BBCube, when more layers are aggressively stacked. Results for 9 (8 + 1), 17 (16 + 1), and 33 (32 + 1) stacked layers are illustrated. Even in the case of 33 layers, BBCube with 3D redundancy indicates superior yield compared with conventional technology at higher yield region.

This result shows that the WOW process with 3D redundancy provides better productivity than the KGD stacking case, even for future applications.

### 11.2.2. Discussion

In both cases of 2D redundancy and 3D redundancy, excellent yield can be realized if we prepare a certain memory cell area. In 3D redundancy, freedom of circuit block replacement, which is orthogonal to 2D redundancy, is provided. Therefore, it is possible to define redundancy success rate "digits" by the "digits" of TSVs, which leads to a higher total number of combinations. Superficially, it looks less productive to add extra wafers for redundancy purposes. By replacing banks within a set of stacked bank groups and introducing sophisticated vertical bank group allocation to realize quasi layer-by-layer operation, the orthogonality of the "digit" by "digit" basis become clear. These 3D redundancy techniques appear to be more practical, and rather rational.

### *11.3. Conclusion of 3D Redundancy*

The excellent performance of BBCube due to the WOW technology and the application of 3D redundancy to utilize wafer form manufacturing have been presented in this chapter. Wafer-on-wafer fabrication was realized with the support of a 3D redundancy scheme, which led us to conclude that BBCube could be the next system scaling enabler.

## 12. Thermal Resistance Comparison of BBCube and Micro-Bumps

In 3D stacking technology, thermal management problems become more difficult due to the vertical thermal resistance of interconnection layers and back end of line (BEOL) [107–110]. Therefore, the temperature of stacked dies increase when they contain more IC chips [111]. Recently, a bumpless 3D multi-stack process using ultra-thin technology was proposed [45–48]. This approach is expected to decrease the vertical thermal resistance. Hence, the total thermal resistance of 3D stacked ICs with and without solder bumps was estimated.

### *12.1. Thermal Resistance Calculation Method*

Figure 54 shows an example of 3D stacked ICs. This structure consists of a Si substrate, Si with TSVs, back end of line (BEOL), vertical interconnections (micro-bumps), and direct contact by TSVs (bumpless). The thermal conductivity of the vertical interconnection was calculated using the Finite Element Method (FEM), and then thermal resistance was calculated using that thermal conductivity. Additionally, the total thermal resistance was calculated using a thermal network method.

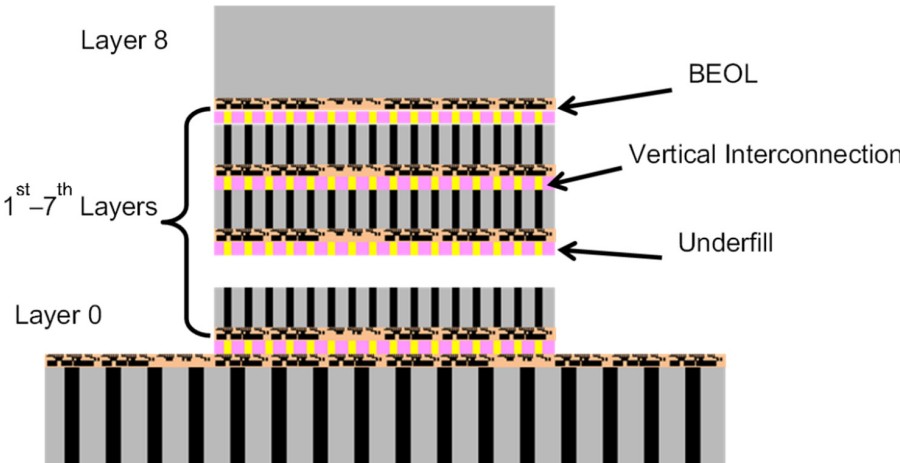

**Figure 54.** A typical 3D stacked memory structure consists of base layer (layer 0) and memory dies (layer 1st to layer 8th). Each die has BEOL layer and vertical interconnect layer.

The temperature rise calculation had four primary steps:

i.    Make assumptions about the IC stack structure,
ii.   Estimate the effective thermal conductivity of each layer, and
iii.  Calculate thermal resistance of each layer, and
iv.   Calculate the temperature rise using the thermal network method.

Figure 55 provides a structural comparison of the micro-bump and bumpless types for IC stacks with 8 layers.

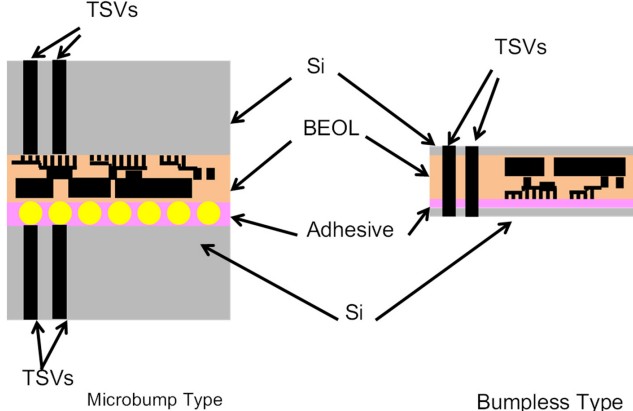

**Figure 55.** A comparison of bump and bumpless interconnects using TSVs for 3D logic/memory stack structures. Each structure consists of a Si layer, a BEOL layer, an interconnection layer, and TSVs. As for the bumpless structure, TSVs are fully transfixed from the top layer to bottom layer.

*12.2. Thermal Resistance of Micro-Bump Vertical Interconnection*

When thermal conductivity of each layer is calculated, the total thermal resistance of the micro-bump 3D stacked ICs can be calculated. The thermal conductivities of micro-bump interconnect were reported by Matsumoto et al. [111]. Additionally, 148, 160.5, and 1.44 W/m/K for Si, Si with TSVs, and BEOL were used, respectively. The thermal conductivity of the vertical interconnection using micro-bumps depends on the bump size, bump pitch, and underfill. Additionally, FEM was used to calculate the thermal performance of micro-bumps. Figure 56 shows the FEM models, and Figure 57 shows the calculation result using the bump occupancy definition shown in Figure 58. From the result, we can see that the use of underfill material is advantageous only when the TSV occupancy is less than 0.05. Using these material thermal conductivities, the thermal resistance of each layer was calculated, and then the total thermal resistance was calculated. In this calculation, the size of the micro-bump was 25 μm and the micro-bump pitch was 50 μm. The total thermal resistance was 1.54 Kcm$^2$/W, which confirms that the thermal resistance of the BEOL and interconnection is too large to reduce the temperature rise.

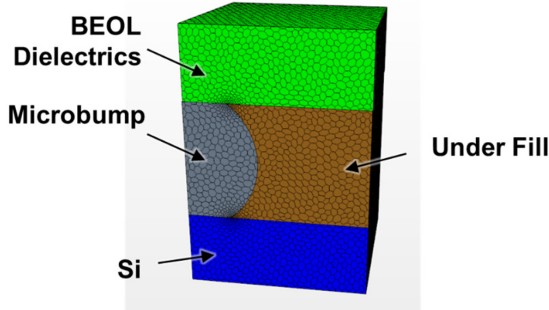

**Figure 56.** Schematic diagram of thermal conductivity of microbump type calculated using a Finite Elements Method Model (FEM model). In this FEM model, the BEOL layer, Si layer, Microbumps, and underfill are modeled.

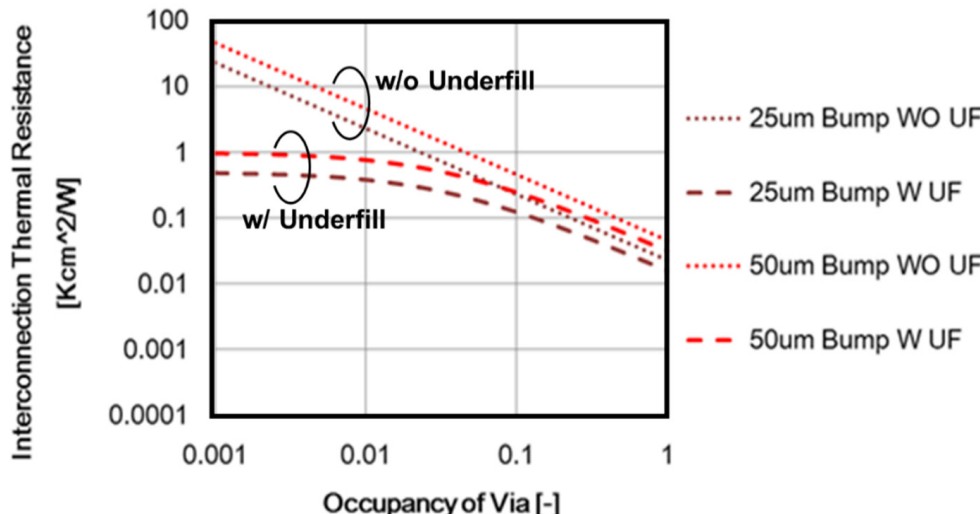

**Figure 57.** Interconnection thermal resistance as a function of TSV occupancy. The definition of TSV occupancy is shown in Figure 58. Two different microbump sizes (25 μm and 50 μm) and the impact of underfill are calculated, where W UF = with underfill material and WO UF = without underfill material. In the case of a 25 μm microbump with and without underfill, thermal resistance is smaller than that of 50 μm. As for the underfill, the thermal resistance with underfill is small compared to no underfill, especially at low occupancy of <0.01.

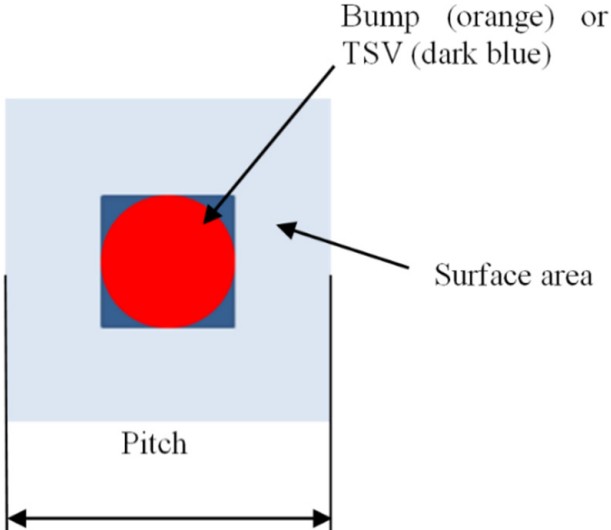

**Figure 58.** The definition of TSV occupancy. TSV occupancy is defined by the area of a bump of TSV divided by the surface area.

### 12.3. Thermal Resistance of BBC

The thermal conductivity of the BEOL with Cu TSV interconnections was calculated using the FEM. Figure 59a shows the FEM model for the BEOL and interconnection in bumpless IC stacks. Figure 59b compares the interconnection thermal resistance for the micro-bump and bumpless types. The thermal resistance for the bumpless type was two orders of magnitude lower than that for the conventional structure. This suggests that only 1% of the total metal area of bumpless TSVs is required to achieve the same thermal resistance, in comparison with the conventional structure. Table 4 shows the thermal resistance results for both types of IC stacks.

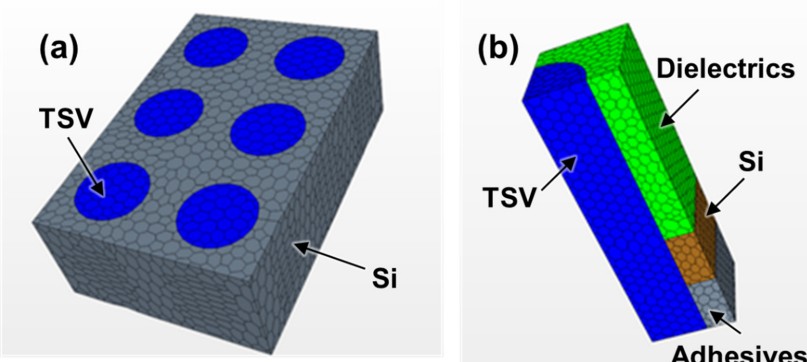

**Figure 59.** Schematic diagram of thermal conductivity of (**a**) bumpless structure calculated using FEM model, where Si layer and TSVs are molded; (**b**) TSV interconnects with microbumps, where the dielectrics (BEOL) layer, Si layer, adhesive layer, and TSV are modeled.

**Table 4.** Total thermal resistance of TSV with microbump and bumpless TSV.

| | | | TSV with Micro Bump 51.4 μm Pitch 25 μm Bump | | Bumpless TSV 512 × 16 TSV 5 μm Gap | |
|---|---|---|---|---|---|---|
| | | Components | Equivalent Thermal Conductivity (W/mK) | Thickness (μm) | Thermal Resistance (Kcm$^2$/W) | Thickness (μm) | Thermal Resistance (Kcm$^2$/W) |
| **DRAM** | Top Chip Rth1 | Si | 148 | 150 | 0.049 | 5 | 0.00034 |
| | | BEOL | 1.44 3.99 | 15 - | 0.104 - | - 15 | - 0.038 |
| | | Interconnection Micro Bump | 2.54 | 20 | 0.079 | - | - |
| | | Interconnection Bumpless | 2.56 | - | - | 5 | 0.02 |
| | 2–8 Chip 7 Layers Rth2–8 | Si with TSV | 160.5 | 50 | 0.003 | 5 | 0.00031 |
| | | BEOL | 1.44 3.99 | 15 - | 0.104 - | - 15 | - 0.038 |
| | | Interconnection Micro Bump | 2.54 | 20 | 0.079 | - | - |
| | | Interconnection Bumpless | 2.56 | - | - | 5 | 0.02 |
| **Logic RthL** | | Si with TSV | 160.5 | 50 | 0.003 | 5 | 0.0003 |
| **Total Thermal Resistance** | | Rth1 + 7 × Rth2–8 + RthL | - | - | **1.54** | - | **0.46** |

As shown in Figure 59, the bumpless process is a kind of via last process, and the TSVs fully go through from bottom to top. In this case, TSVs are formed by copper and their thermal conductivity is around 400 (W/m/K). This value is around 280 times larger than BEOL thermal conductivity and around 160 times larger than microbump interconnect thermal conductivity. Thus, only a 1% volume fraction is effective for effective thermal conductivity improvement. In addition, the bumpless interconnection thickness is 4 times thinner than microbump one, hence the thermal resistance of interconnection is more than 4 times smaller. The interconnection and BEOL thermal resistances for the bumpless type were almost 4 and 3 times smaller, respectively, than those for the conventional structure.

*12.4. Temperature Rise Calculation Result*

The temperature rise for each layer was calculated using:

$$T_M = \sum_{k=1}^{M} \left( R_k * \sum_{l=k}^{M} Q_{N-l+1} \right) \tag{10}$$

where,

$$T_M : \text{Temperature rise of layer M} \left( {}^\circ C \right)$$

$$R_k \ : \text{Thermal Resistance of layer k} \ \left( \frac{\text{Kcm}^2}{\text{W}} \right)$$

$$Q_k \ : \text{Heat Generation of layer k (W)}$$

The temperature rise was caused by the product of its own thermal resistance and the heat generated by the layer below it.

Figure 60 shows that the temperature increased as a function of the number of DRAM dies, and a comparison for the micro-bump and bumpless types. "Layer x" represents the DRAM dies. The maximum temperature rise $\Delta T$ for the no microbump case (BBCube) was around 5.8 °C, which is almost one-fourth that of the microbump case.

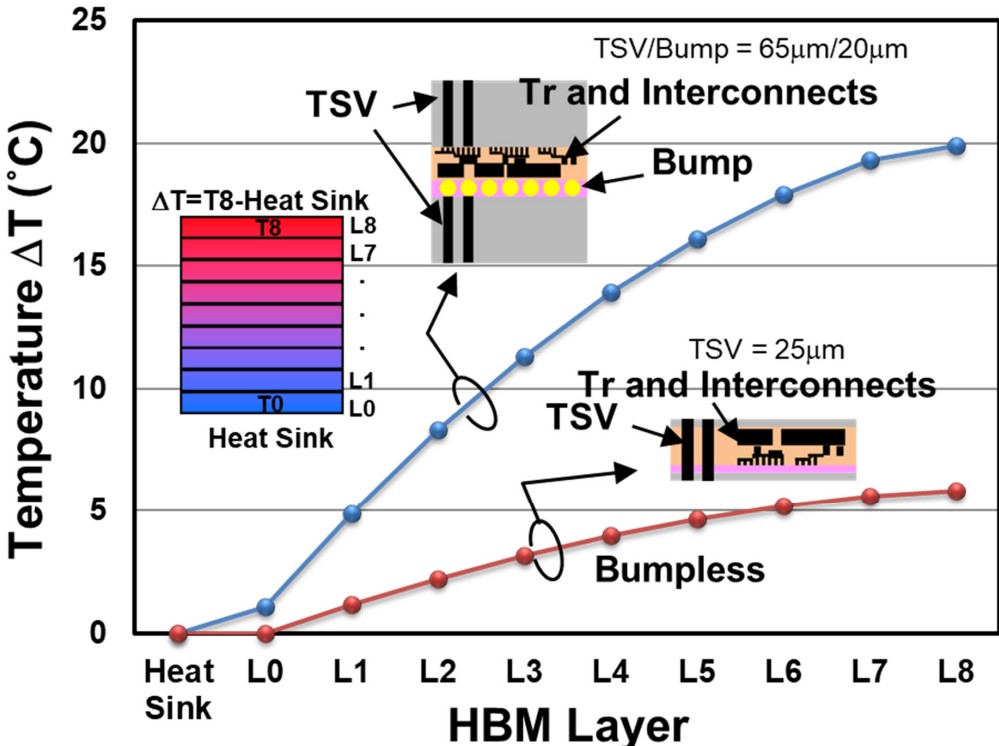

**Figure 60.** Temperature rises with and without microbumps as a function of the number of stacked dies. "Layer x" represents the DRAM dies. The maximum temperature rise for the no microbump case (BBCube) is around 5.8 °C, which is lower than that of the microbump case, i.e., 20 °C.

*12.5. Thermal Resistance Comparison Conclusion*

We established a calculation method for evaluating the thermal resistance of 3D stacked ICs following the method in Matsumoto et al. [111]. Using this method, we calculated the temperature rise for each layer in 3D IC stacks with both microbump and bumpless vertical interconnections. For the ICs modeled in this study, the total thermal resistance of organic layers 20 and 5 μm thick were 1.54 and 0.46 Kcm$^2$/W, respectively.

## 13. Summary and Conclusions

Due to the demand for post-scaling in device structures, three-dimensional integration technologies are expected to be increasingly employed. By doing so, when wafers with micrometer thickness are stacked, the total thickness is reduced, and the transistor capacity increases in proportion to the number of wafers. Increasing the TSV interconnects density enables terabyte-level bandwidth without sacrificing energy efficiency. Power consumption and heat dissipation are especially important for high-density modules, such as 2.5D and 3D systems. 2.5D, which is not a physical term, refers to a high-speed, high-bandwidth system that incorporates and integrates a three-dimensional memory such as HBM, GPU

(Graphic Processing Unit), and MPU on an interposer, and is a general term for the back-end processes. In recent years, it has become a product differential technology to combine multiple chips and passive components with different functions into one system module. The authors' research organization, the "WOW Alliance", has proposed the BBCube architecture using WOW and COW processes for 2.5D and 3D systems including passive devices, as described in this paper.

As the number of stacked wafers increases, the number of incoming wafers in manufacturing increases proportionally [112]. Recently, volume production with 80,000 wafers per month has been used. To maintain the same throughput with stacks of 8 DRAM wafers, the number of incoming wafers will be 640,000 per month. Without considering facility costs and running costs, it would be possible to increase the size of fabrication plants. However, a production line with eight-times larger footprint may not balance the production costs. Thus, in the future, enlarged wafer size or an alternative approach such as a combination of reducing total process steps with very high throughput may be reconsidered in this situation.

If the alignment accuracy of wafer stacking is improved, about 1 to 10 million TSVs can be formed per square centimeter. Such large-scale I/O is too high for DRAM stacking, but if scaling down of TSVs and layout flexibility evolve, it will be possible to stack MPU logic and SRAM cache memory individually. If the power distribution and ground can be located directly beneath of SRAM cell, stable current and low applied voltage <0.7 V with low noise can be realized because they can be connected with equivalent lengths and high parallelity by micrometer-level short interconnects. Such high-density TSV interconnects in conjunction with BBCube (low power consumption) will help to reduce the excess heat of 3D systems.

In summary, it is possible to achieve the next step in the semiconductor roadmap by employing three-dimensional integration technology, as discussed. Although it is necessary to develop 3DI technology with high productivity, such as front-end wafer technology, many of those mature processes can be applied. Thus, the new technology for 3DI is only the thinning and stacking processes. These technologies can also be improved as there are well-known technologies from the front-end and novel material candidates, which are expected to become mature by applying the know-how gained over many years in the semiconductor industry.

**Author Contributions:** Conceptualization, T.O.; BBCube WOW and COW processes, T.O.; BBCube memory design, K.S. and N.C.; BBCube memory redundancy, S.S.; BBCube thermal analysis, H.R.; writing—review and editing, Sections 1–6 and 13, T.O., Sections 7–10, K.S. and N.C.; Section 11, S.S., Section 12, H.R.; supervision, T.O. All authors have read and agreed to the published version of the manuscript.

**Funding:** This research was funded by the WOW Alliance of Tokyo Institution of Technology (Tokyo, Japan).

**Acknowledgments:** This study was carried out based on the three-dimensional integration development program of the WOW Alliance at the Tokyo Institute of Technology, and the authors thank the alliance members of more than 30 companies for their cooperation.

**Conflicts of Interest:** The authors declare no conflict of interest.

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
