# Peer review of "Review of Bumpless Build Cube (BBCube) Using Wafer-on-Wafer (WOW) and Chip-on-Wafer (COW) for Tera-Scale Three-Dimensional Integration (3DI)"

_electronics, doi:10.3390/electronics11020236_

Round 1

Reviewer 1 Report

There are several things which must be improved before it can reach merits of publications:

Comment:

  1. A good review paper is supposed to review the relevant previous literature comprehensively and to provide insights based on them. In this manuscript, many previous works were simply summarized with no in-depth discussions or insights from the authors’ point of view.

  1. The author needs to explain why the refresh time can be improved by increasing the thickness of the backside defects layer using grinding @line111.

  1. In terms of Competitive BBC DRAM Structure, this paper proposes a novel scheduling strategy named four-phase shielded I/O scheme”,Please refer to Figure 8-12 to elaborate the details of this scheduling strategy and illustrate the great advantages this scheduling strategy brings to DRAM access with the BBCube architecture.

  1. In the section about BBCube NAND, the article mentioned Limitations of Stacked WL Tiers in 3D NAND Chip,please analyze the limitations of this architecture based on specific examples. Detailed examples and explanations about WL Tiers architecture are missing here.

  1. The figure in page 44 should be "Figure 11-8", instead of "Figure 10-8".

  1. More details about "Quasi layer-by-layer Operation" should be explained @line785. Reader might be confused about this technique's benefits.

  1. In figure 12-6, each part represented by each color should be marked like this figure 12-3.

  1. The author should explain the deep reason why the thermal resistance for the bumpless type was much lower than that for the Micro-Bump structure.

Author Response

I have attached a response to the peer review check. Thanks.

Reviewer 2 Report

  • The authors present an interesting wafer on wafer and chip on wafer integration methodology and the paper overall is well written
  • Unfortunately the manuscript is too long, so suggest the authors split this manuscript into 2 separate papers if possible, one that talk about materials, processing, and characterization and the other talking about the larger architecture, roadmaps, thermal and yield considerations
  • Please address the following comments
    • How does this method compare in terms of performance with hybrid bonding of wafer to wafer or chip to wafer? expect both to be similar, can the authors explain the motivation of using permanent adhesive as opposed to hybrid bonding method?
    • fig 3-2 caption need correction '9 to 1um' not mm
    • fig 3-2 electrical characterization of nmos and pmos graphs are not clear, suggest including a higher resolution image or removing the plots
    • line 113 - explain the source of copper to cause contamination post Si backgrind and cmp
    • fig 3-5 are the 4 plots from a prior publication or course or workshop? if yes, suggest removing the plots and instead just providing a reference to the material
    • fig 4-2 bevel angle vs defect have already been reported in ref 25, suggest removing this plot and providing reference to the material
    • fig 4-2 what is the thickness of the si for the various bevel angles?
    • fig 4-2 a) typo 'bebel' I think it should be 'bevel'
    • line 296 - are the weight loss measurements made on non-bonded sample? 
    • fig 4-3 can the authors share CSAM data to show no voiding at the interface after curing the permanent adhesive? Also CSAM data after 1000 cycles will be useful
    • What is the temperature range for cycling?
    • What is the thickness of the sidewall passivation? is it SiO2?
    • Explain how the permanent adhesive if etched post si etch
    • Is the leakage current measurement between isolated TSV pads or between TSV and substrate? is the substrate grounded in the measurements?
    • fig 5-9 was the on-wafer measurement a single point measurement? Can you include the sweep data similar to on interposer?

Author Response

(The authors gave the same response as above.)

Round 2

Reviewer 2 Report

  • Thank you for addressing the questions from the first review
  • Suggest a complete proof read and minor corrections to english language to make this a high quality scientific paper

Author Response

This manuscript has been edited by a native-English-speaking science editor.
